# Topological Dirac nodal lines and surface charges in fcc alkaline earth metals

Motoaki Hirayama[1,2], Ryo Okugawa[1], Takashi Miyake[3] & Shuichi Murakami[1,2]

In nodal-line semimetals, the gaps close along loops in $\mathbf{k}$ space, which are not at high-symmetry points. Typical mechanisms for the emergence of nodal lines involve mirror symmetry and the $\pi$ Berry phase. Here we show via *ab initio* calculations that fcc calcium (Ca), strontium (Sr) and ytterbium (Yb) have topological nodal lines with the $\pi$ Berry phase near the Fermi level, when spin–orbit interaction is neglected. In particular, Ca becomes a nodal-line semimetal at high pressure. Owing to nodal lines, the Zak phase becomes either $\pi$ or 0, depending on the wavevector $\mathbf{k}$, and the $\pi$ Zak phase leads to surface polarization charge. Carriers eventually screen it, leaving behind large surface dipoles. In materials with nodal lines, both the large surface polarization charge and the emergent drumhead surface states enhance Rashba splitting when heavy adatoms are present, as we have shown to occur in Bi/Sr(111) and in Bi/Ag(111).

[1] Department of Physics, Tokyo Institute of Technology, Ookayama, Meguro-ku, Tokyo 152-8551, Japan. [2] TIES, Tokyo Institute of Technology, Ookayama, Meguro-ku, Tokyo 152-8551, Japan. [3] Research Center for Computational Design of Advanced Functional Materials, AIST, Tsukuba 305-8568, Japan. Correspondence and requests for materials should be addressed to S.M. (email: murakami@stat.phys.titech.ac.jp).

Recent discoveries of topological semimetals have taught us that the **k**-space topological structure of electronic bands plays a vital role in a number of materials. This class of topological semimetals includes Weyl semimetals[1,2], Dirac semimetals[3,4] and nodal-line semimetals (NLSs)[5–15]. In topological semimetals, the conduction and valence bands touch each other at some generic points (as in Dirac and Weyl semimetals) or along curves (in NLSs) in **k**-space. Such degeneracies do not originate from high-dimensional irreducible representations at such **k** points, but rather from the interplay between **k**-space topology and symmetry. Dirac semimetals have been realised in $Na_3Bi$ (refs 16,17) and $Cd_3As_2$ (refs 18,19). Weyl semimetals have been proposed to exist in pyrochlore iridates[2], $HgCr_2Se_4$ (ref. 20), Te under pressure[21], $LaBi_{1-x}Sb_xTe_3$, $LuBi_{1-x}Sb_xTe_3$ (ref. 22), transition-metal dichalcogenides[23,24] and $SrSi_2$ (ref. 25). Consistent with theoretical predictions[26,27], the TaAs class of materials has been experimentally found to be Weyl semimetals[28–32].

In the present study we focus on NLSs. Two typical origins of the nodal lines are (A) mirror symmetry and (B) the $\pi$ Berry phase, as explained in the Methods section. To our knowledge, proposals for NLSs have thus far been restricted to the former mechanism. Dirac NLSs having Kramers degeneracy include carbon allotropes[7,8], $Cu_3PdN$ (refs 9,10), $Ca_3P_2$ (refs 11,12), $LaN$[13], $CaAgX(X = P,As)$[14] and compressed black phosphorus[15]; similarly, Weyl NLSs, which has no Kramers degeneracy, include $HgCr_2Se_4$ (ref. 20) and $TlTaSe_2$ (ref. 33). The latter mechanism is also found to occur in some of these materials; that is, the nodal line survives despite external disruption of mirror symmetry. Thus far, no purely topological NLSs resulting from the latter mechanism have been proposed.

In the present study we propose on the basis of *ab initio* calculation that the alkaline-earth metals Ca, Sr and Yb have topological nodal lines when the spin–orbit interaction (SOI) is neglected. In reality, the SOI is nonzero, especially for Yb, giving rise to a small gap along the otherwise gapless nodal lines. In fact, the existence of nodal lines has been observed[34–36] and has been used to explain resistivity data. Nevertheless, its topological origin and its relationship with surface states remain unexplored. Here we show their physical origin. We also calculate the Zak phase along some reciprocal lattice vector and show that the Zak phase is either $\pi$ or 0, depending on the momentum regions divided by the nodal lines. As the Zak phase is related to polarization, the region with the $\pi$ Zak phase gives rise to a polarization charge at the surface normal to the reciprocal lattice vector. We show that, contrary to common belief, the $0/\pi$ Zak phase is not related to the absence or presence of surface states. Unlike insulators, carriers screen this surface polarization charge from the $\pi$ Zak phase, leaving behind surface dipoles. Finally, we expect the large surface dipoles due to nodal lines to enhance surface Rashba splitting, possibly contributing to Rashba splitting in Bi/Sr(111) (Rashba energy: $E_R \sim 100$ meV). The large Rashba splitting ($E_R \sim 200$ meV) on the surface of Bi/Ag(111)[37] is also attributed to hybridization between the Bi states with emergent surface states from the nodal lines in Ag. Thus, the nodal lines are shown to enhance surface Rashba splitting, which is potentially important for spintronics applications.

## Results

### Band structures of fcc alkaline earth metals.
Ca, Sr and Yb are non-magnetic metals, having a face-centred cubic (fcc) lattice with lattice parameter $a$ (Fig. 1a). The space group of fcc is $Fm\bar{3}m$ (No. 225). At higher pressure, interesting phase transitions have been observed in these metals. In Yb, the metal-insulator transition occurs at 1.2 GPa[38]. Ca and Sr also exhibit semimetallic

behaviour under pressure[39,40]. The first structural transition from fcc to body-centred cubic takes place in at 19–20 GPa[41]. High-temperature superconductivity is observed in Ca at 29 and at 216 GPa after several structural transitions[42].

We determine their electronic structures by *ab initio* calculations, as explained in the Methods section. Figure 2a shows the electronic structure of Ca obtained by local density approximation (LDA). The Brillouin zone is shown as a truncated octahedron in Fig. 2b. In a Ca atom, a gap exists between the fully occupied $4s$ orbitals and unoccupied $3d$ and $4p$ orbitals. When the atoms form a crystal, these orbitals form bands with a narrow gap or pseudo-gap near the Fermi level. The top of the valence band, which is relatively flat near the $L$ points, originates from the $p$ orbital oriented along the [111]-axis having strong $\sigma$ bonding, whereas most of the other valence bands originate from the $s$ and $d$ orbitals. Around the $L$ points, the relatively flat valence band crosses the dispersive conduction band. It produces four nodal lines around the $L$ points within approximately $\pm 0.01$ eV near the Fermi level, as shown in Fig. 2b. (There appears to be eight nodal lines, but the nodal lines in the same colour are identical.) The four nodal lines are mutually related by $C_4$ symmetry and are oriented slightly away from the faces of the first Brillouin zone, except for the points along the $L–W$ lines ($Q_1$ in Fig. 2c) because of $C_2$ symmetry. The nodal lines do not lie on mirror planes; therefore, they do not arise from mirror symmetry. Because of the topology resulting from the $\pi$ Berry phase (see Methods and Supplementary Note 1), closing of the gap is not limited to the $Q_1$ points on the $L–W$ lines, instead extending to form nodal lines. Indeed, we numerically confirmed that the Berry phase around each nodal line is $\pi$.

There are other ways of topological characterization of the nodal lines, distinct from the Berry phase. For example, one way of topological characterization is the $\mathbb{Z}_2$ indices defined in ref. 9, calculated as products of the parity eigenvalues of the valence bands at the time-reversal invariant momenta. We find that all $\mathbb{Z}_2$ indices are even (trivial) for Ca, Sr and Yb. The existence of the nodal lines is consistent with the trivial $\mathbb{Z}_2$ indices, as the number of nodal lines between the time-reversal invariant momenta is even. In that sense, these metals can be called 'weak' NLSs, in

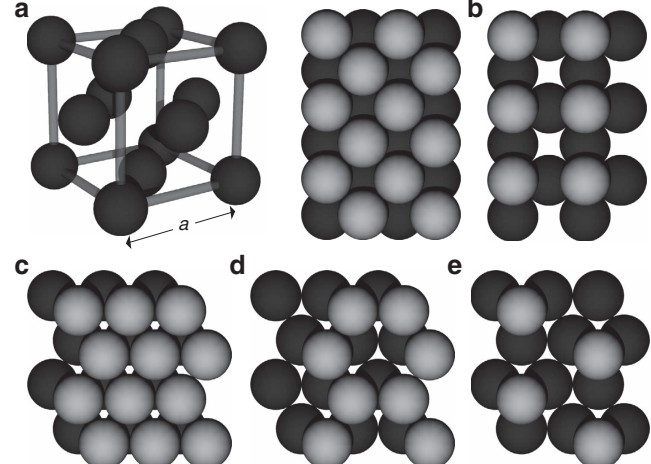

**Figure 1 | Bulk and surface structure.** (**a**) Crystal structure of fcc Ca, Sr and Yb, and that of the (001) surface (black circles) with surface atoms (grey circles). (**b**) The same surface orientation but with half of the atoms per unit cell on the surface. (**c**) Crystal structure of the (111) surface (black circles) with surface atoms (grey circles). (**d,e**) The same surface orientation but with two-thirds and one-third, respectively, of the atoms per unit cell on the surface.

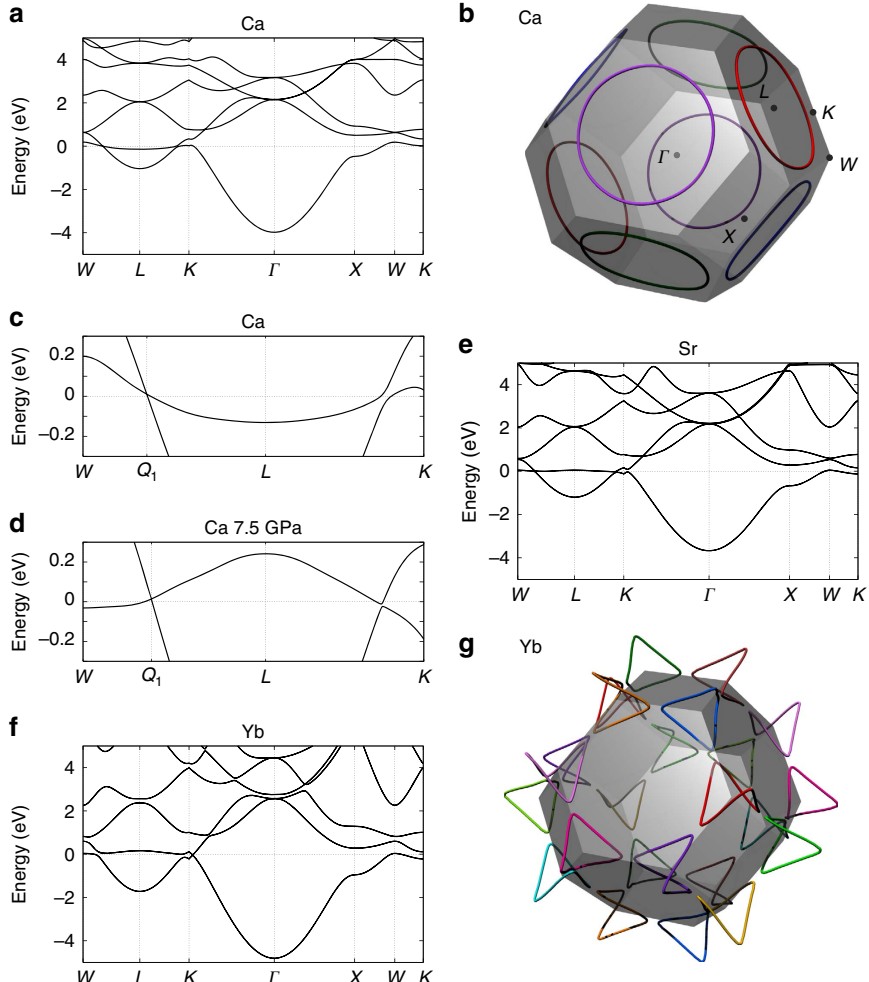

**Figure 2 | Electronic band structure and nodal line. (a)** Electronic band structure of Ca in the LDA. **(b)** Nodal lines and the Brillouin zone of Ca, where identical nodal lines (modulo reciprocal vector) are shown in the same colour. **(c,d)** The magnified electronic band structure of Ca at ambient pressure and at 7.5 GPa, respectively, in the LDA. **(e,f)** The electronic band structures of Sr and Yb, respectively, in the LDA + SO. **(g)** The nodal lines of Yb in the LDA. The energy is measured from the Fermi level.

analogy with weak topological insulators; namely, the existence of nodal lines does not arise from bulk $\mathbb{Z}_2$ indices. Another $\mathbb{Z}_2$ index is defined for each nodal line in ref. 6. If it is non-trivial, then it prevents the nodal line from disappearing by itself. In the present case of Ca, this $\mathbb{Z}_2$ index is trivial, as shown in detail in the Supplementary Note 4.

At ambient pressure, Ca is not an NLS, because the two bands forming the nodal lines both disperse downward around the $L$ points (Fig. 2c). Meanwhile, Ca becomes an NLS under pressure, as shown in the band structure at 7.5 GPa in Fig. 2d; a similar conclusion has been reached in previous works[34,35] without showing the topological origin of the nodal lines. Here, the pressure increases the energy of the $p$ orbital relative to that of the $s$ and $d$ orbitals.

The electronic structure of Sr in the LDA with relativistic effect (LDA + SO, see Methods) is shown in Fig. 2e. The SOI is not strong over the entire $\mathbf{k}$ space. The band at the $L$ points near the Fermi level is relatively flat compared with that in Ca, because the energy difference between the $5s$, $4d$ and $5p$ orbitals is larger than that between the $4s$, $3d$ and $4p$ orbitals. With the SOI neglected, the four nodal lines occur around the $L$ points, as is the case with Ca. The nodal line is fully gapped with the SOI; for example, the degeneracy on the $L$–$W$ line splits by $\sim 0.04$ eV because of the SOI.

Figure 2f shows the electronic structure of Yb in the LDA + SO. The energy splitting on the $L$–$W$ line reaches

$\sim 0.2$ eV. Unlike Ca and Sr, the nodal lines without the SOI in Yb are qualitatively different from those in Ca and Sr; hybridization between four nodal lines around the $X$ points causes a Lifshitz transition, that is, a recombination of the nodal lines, and 12 small nodal lines appear around the $W$ points (Fig. 2g). A similar recombination of nodal lines is seen when the lattice constant of the Ca crystal is increased in the numerical calculation, which gradually reduces the crystal to the atomic limit. The nodal lines form around the $W$ points after recombination, subsequently shrinking toward the $W$ points and disappearing. Compared with Ca and Sr, Yb is consequently closer to band inversion between the atomic limit and the crystal.

**Zak phase and surface states**. We now show the surface states of Ca, Sr and Yb. Figure 3a shows the electronic structure of the Ca(110) surface and Fig. 3b shows its charge distribution at the Fermi level. Similar surface states are also found in the Sr(110) and Yb(110) surfaces (Fig. 3c,d). Surface states connecting the gapless points exist near the Fermi level around the $\bar{X}$ point, isolated from the bulk states. In particular, whereas the SOI opens a small gap at the nodal lines in Sr and Yb, the surface states persist by continuity because of nodal lines. Out of the four nodal lines in this (110) surface, two overlap each other, resulting in a projected nodal line around the $\bar{Z}$ point. The other two nodal

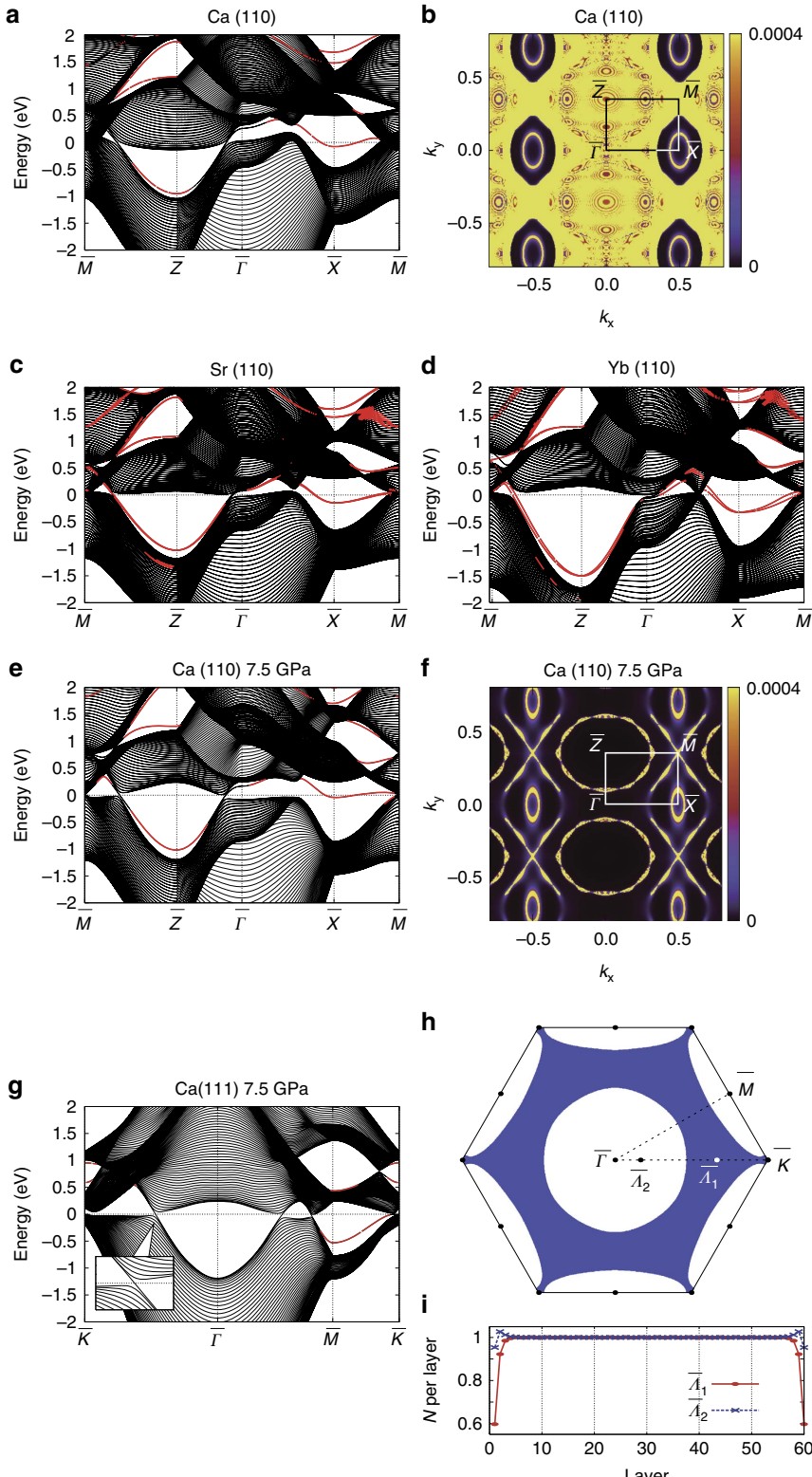

**Figure 3 | Topological surface state and surface polarization charge.** (**a**) Electronic band structure for the the(110) surface of Ca in the LDA. The symmetry points are $\bar{\Gamma} = (0, 0, 0)$, $\bar{X} = (\pi/a)(\sqrt{2}, 0, 0)$, $\bar{Z} = (\pi/a)(0, 0, 1)$ and $\bar{M} = (\pi/a)(\sqrt{2}, 0, 1)$. (**b**) Intensity colour plot of the charge distribution in **k** space at the Fermi level within one atomic layer near the surface (the unit for the **k** vector is $2\sqrt{2}\pi/a$). (**c,d**) Electronic band structures of Sr and Yb for the (110) surface in the LDA + SO, respectively. (**e,f**) Results for Ca at 7.5 GPa corresponding to parts **a,b**. (**g**) Electronic band structure for the (111) surface of Ca in the LDA, with states close to the nodal line magnified in the inset. (**h**) Dependence of the Zak phase on the surface momentum $\mathbf{k}_{\parallel}$. The shaded region represents $\mathbf{k}_{\parallel}$ with the $\pi$ Zak phase, whereas other regions represent that with the 0 Zak phase. (**i**) Charge profile for two values of the surface momentum $\mathbf{k}_{\parallel} = \bar{\Lambda}_1, \bar{\Lambda}_2$ in real space over the thickness direction of the slab. The vertical axis represents the charge density per surface unit cell within each atomic layer, measured in units of electronic charge, $-e$. In parts **a,c–e,g**, wavefunctions monotonically decreasing toward the bulk region are shown in red. The energy is measured from the Fermi level.

lines nearly become segments crossing each other at the $\bar{M}$ point. Figure 3e is the electronic structure of the Ca(110) surface at 7.5 GPa. It is in the NLS phase and the states at the Fermi level (Fig. 3f) consist almost exclusively of the surface states.

We calculate the Zak phase (the Berry phase), which is an integral of the Berry connection of the bulk wavefunction along a certain reciprocal vector $\mathbf{G}$. For the calculation, we decompose the wavevector $\mathbf{k}$ into the components along $\mathbf{n} \equiv \mathbf{G}/|\mathbf{G}|$ and perpendicular to $\mathbf{n}$: $\mathbf{k} = k_\perp \mathbf{n} + \mathbf{k}_\parallel$, $\mathbf{k}_\parallel \perp \mathbf{n}$. The integral with respect to $k\perp$ is calculated with fixed $\mathbf{k}_\parallel$. The Zak phase is defined in terms of modulo $2\pi$ because of the gauge degree of freedom. We focus on the cases without the SOI and neglect the spin degeneracy. As discussed in a previous work[43], the Zak phase is related to charge polarization at surface momentum $\mathbf{k}_\parallel$ for a surface perpendicular to $\mathbf{n}$ (see the Methods section for details). For example, in a one-dimensional insulating system, the product of the Zak phase and $e/(2\pi)$ is equal to the polarization, that is, the amount of surface polarization charge, modulo $e$[43]. In addition, in a three-dimensional system, if the system at each $\mathbf{k}_\parallel$ is regarded as a one-dimensional system, then the product of the Zak phase $\theta(\mathbf{k}_\parallel)$ and $e/(2\pi)$ is equal to $\sigma(\mathbf{k}_\parallel)$ modulo $e$, where $\sigma(\mathbf{k}_\parallel)$ is the amount of surface charge for the one-dimensional system at given $\mathbf{k}_\parallel$. For an insulator, a surface polarization charge density $\sigma_{\text{total}}$ at the given surface is given by $\sigma_{\text{total}} = \int \frac{\mathrm{d}^2 k_\parallel}{(2\pi)^2} \sigma(\mathbf{k}_\parallel)$ (ref. 43). As the Berry phase around the nodal line is $\pi$, the Zak phase jumps by $\pi$ as $\mathbf{k}_\parallel$ changes across the nodal line. This is confirmed in our case. The resulting Zak phase is 0 in the entire $\mathbf{k}_\parallel$ space for the (110) surface, because the two nodal lines out of the four overlap each other, while the other two nearly becomes segments. In the (001) surface, the Zak phase is also 0 everywhere, because four nodal lines overlap each other in two pairs and the Zak phase is doubled. Meanwhile, the Zak phase for the (111) surface (Fig. 3g) is $\pi$ outside of the nodal line, as shown as the shaded region in Fig. 3. Within this $\mathbf{k}_\parallel$ region, $\sigma(\mathbf{k}_\parallel)$ takes a value $\sigma(\mathbf{k}_\parallel) \equiv e/2 \pmod{e}$[43], inevitably leading to a surface polarization charge. When the surface termination is fixed, the value of $\sigma(\mathbf{k}_\parallel)$ is determined without the indeterminacy modulo $e$.

For example, the Zak phases for points $\bar{\Lambda}_1$ and $\bar{\Lambda}_2$ in Fig. 3h are $\pi$ and 0, respectively, and the surface polarization charges $\sigma(\mathbf{k}_\parallel)$ at these wavevectors are $e/2$ and 0 (mod $e$). As surface states exist neither at $\bar{\Lambda}_1$ nor at $\bar{\Lambda}_2$, this difference in surface polarization charges are due to charge distribution of bulk valence bands, as demonstrated in Fig. 3i. At $\bar{\Lambda}_2$ the charge distribution is almost constant even near the surface, whereas at $\bar{\Lambda}_1$ it decreases by $\sim (-e)/2$ near each surface, consistent with the value of $\sigma(\mathbf{k}_\parallel) \equiv e/2 \bmod e$. As the two surfaces of the slab are equivalent because of inversion symmetry, the total charge at $\bar{\Lambda}_1$ is less than that at $\bar{\Lambda}_2$ by one electron (that is, charge $(-e)$). This difference is attributed to one state which traverses the gap from the valence band to the conduction band along the $\bar{K} \rightarrow \bar{\Gamma}$ direction (inset of Fig. 3g; notably, this state is not a surface state, as it disappears at the limit of infinite system size. The small gap between $\bar{K}$ and $\bar{\Gamma}$ is a minigap because of the finite-size effect and this gap goes to zero in the infinite system size.). To summarize, an $e/2$ surface polarization charge from the $\pi$ Zak phase is attributed to bulk states. It holds true even when the Zak phase is not quantized; a non-zero Zak phase $\theta$ implies that the bulk states have excess polarization charge at the surface.

Thus, $\sigma(\mathbf{k}_\parallel)$ takes the value $\sigma(\mathbf{k}_\parallel) = \frac{e}{2}$ in the shaded region in Fig. 3h, the area of which is 0.485 of the total area of the Brillouin zone. Therefore, the surface polarization charge density is $\sigma = 0.485 \cdot \frac{e}{2A_{\text{surface}}} \sim \frac{0.243e}{A_{\text{surface}}}$ where $A_{\text{surface}}$ is the area of the surface unit cell. A non-zero surface polarization charge in this centrosymmetric crystal seems unphysical. Nevertheless, it does

not violate the inversion symmetry, because the amount of surface charge is the same for two surfaces of a slab of finite thickness. Notably, it is a surface polarization charge if we regard the system as a collection of one-dimensional systems for each $\mathbf{k}_\parallel$. In reality, the excess surface charge are screened by free carriers because of the existence of free carriers, as discussed later in this study.

It is commonly believed that when $\theta(\mathbf{k}_\parallel)$ equals $\pi$ at some $\mathbf{k}_\parallel$, the drumhead surface states appear at $\mathbf{k}_\parallel$; this is indeed the case when the system has chiral symmetry, according to a theorem in a previous work[44]. Here, surface states are defined as states having a finite penetration depth in the limit of an infinite system size. Nevertheless, it is not always true in general systems. Comparison between the (111) surface states (Fig. 3g) and the value of the Zak phase (Fig. 3h) shows no direct relationship between the absence or presence of the surface states and the $0/\pi$ Zak phase. It is in fact natural, as shown by the following discussion. Suppose there is a surface state within the gap at $\mathbf{k}_\parallel$ near the nodal line. If it is occupied, it then contributes $(-e)$ to the surface polarization charge; if unoccupied, it does not contribute. Thus, the presence or absence or occupancy of surface states affects the surface polarization charge by an integer multiple of $e$ and this cannot account for $e/2$ surface charge from the $\pi$ Zak phase. Therefore, the $\pi$ Zak phase due to the nodal line does not imply the existence of surface states[6] and the presence or absence of surface states depends on details of the surface being considered.

A previous work has shown that only in systems with chiral symmetry the $\pi$ Zak phase indicates the presence of boundary states (at zero energy)[44]. This is consistent with our conclusion above. When the Zak phase for the bulk occupied bands at certain $\mathbf{k}_\parallel$ value is $\pi$, the surface polarization charge for the occupied bands is $\sigma_{\text{occ.}} \equiv e/2 \pmod{e}$ at this wavevector. From the chiral symmetry, the bulk unoccupied bands also have the same surface polarization charge $\sigma_{\text{unocc.}} = \sigma_{\text{occ.}}$. Therefore, the total surface charge for all of the bulk bands is $\sigma_{\text{unocc.}} + \sigma_{\text{occ.}} = 2\sigma_{\text{occ.}} \equiv e \pmod{2e}$. Thus, there is an odd number of surface states, which accommodate excess electrons at the surface, and there is a zero-energy surface state due to chiral symmetry. Hence, the chiral symmetry is essential to relating the $\pi$ Zak phase to the presence of surface states.

We emphasize that this surface charge is determined from the bulk bands. This can be seen by considering the surfaces with periodic depletion of some atoms, such that the surface forms a superstructure. For the (001) surface, we consider two patterns for the $\sqrt{2} \times \sqrt{2}$ superstructure, as shown in Fig. 1b,c. Figure 1b represents the perfect surface; in Fig. 1c, half of the surface atoms per unit cell are present. For the (111) surface, we consider three patterns for the $\sqrt{3} \times \sqrt{3}$ superstructure, as shown in Fig. 1c–e. Figure 1c represents the perfect surface; in Fig. 1d,e, two-thirds and one-third, respectively, of the surface atoms per unit cell are present. The band structures are shown in Fig. 4a–e. We also found that for the (001) surface, two patterns for the $\sqrt{2} \times \sqrt{2}$ (a,b) give the same Zak phase (see Fig. 4f). Similarly, three patterns for the $\sqrt{3} \times \sqrt{3}$ superstructure for the (111) surface (c–e) give the same Zak phase (see Fig. 4g). Although the Zak phase generally depends on the surface termination via the choice of the unit cell[45], it is unchanged in the present case when the surface termination is changed for a fixed surface orientation, as shown in the Supplementary Note 3. This result is natural, because the Zak phase and resulting surface polarization charge (modulo $e$) are bulk quantities and are independent of formation of surface superstructure.

**Nodal lines and Rashba splitting.** Thus far we have shown that the nodal lines affect the surface in two ways. One is through the surface charge via the Zak phase. Across the nodal lines the Zak phase changes by $\pi$ and regions with $\pi$ Zak phase (having $e/2$ (mod $e$) surface charge) encircled by the nodal lines appear, giving

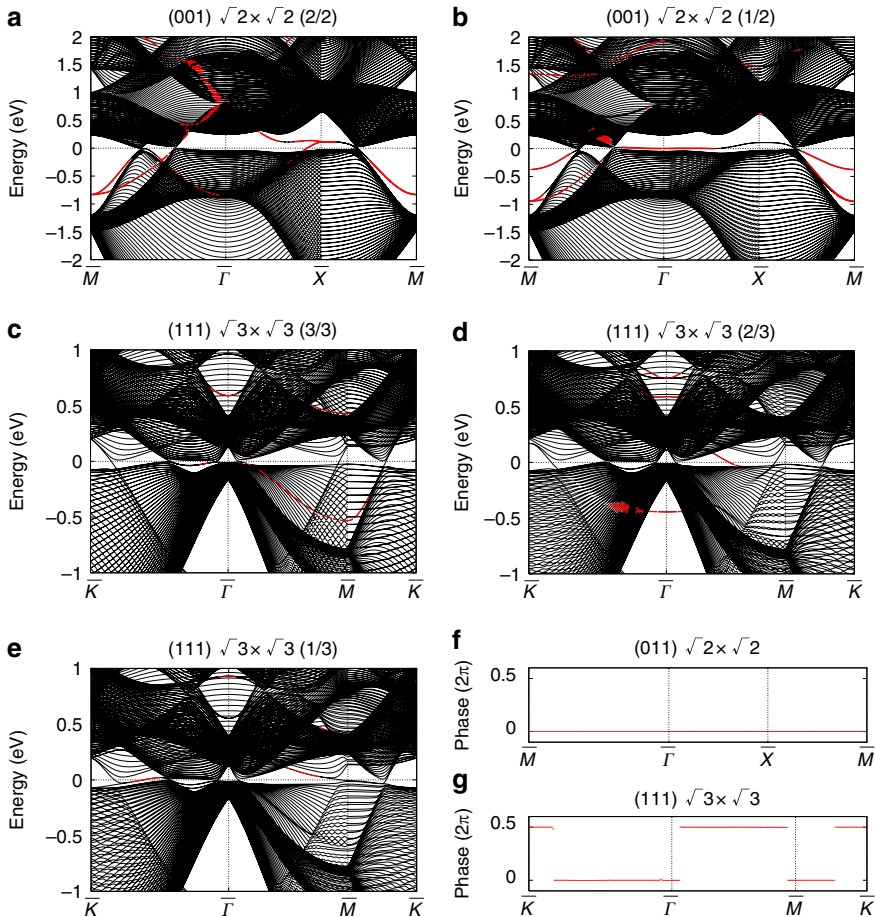

**Figure 4 | Topological surface state on long-range superstructure.** Electronic band structure of Ca at 7.5 GPa, with **a,b** showing long-range $\sqrt{2} \times \sqrt{2}$ structures on the (001) surface and **c–e** showing long-range $\sqrt{3} \times \sqrt{3}$ structures on the (111) surface in the LDA. The surfaces in **a,c** are flat and those in **b–e** have one-half, one-third and two-thirds of the surface atoms per unit cell, respectively (**b–e** correspond to Fig. 1b–e, respectively). The symmetry points in **a,b** are $\bar{\Gamma} = (2\pi/a)(0, 0, 0)$, $\bar{X} = (\pi/a)(1, 0, 0)$ and $\bar{M} = (\pi/a)(1,1, 0)$; those in **c–e** are $\bar{\Gamma} = (2\sqrt{2}\pi/\sqrt{3}a)(0, 0, 0)$, $\bar{M} = (\sqrt{2}\pi/\sqrt{3}a)(0, 1, 0)$ and $\bar{K} = (\sqrt{2}\pi/\sqrt{3}a)(\sqrt{3}, 1, 0)$. Wave functions decreasing monotonically towards the bulk region are shown in red. The energy is measured from the Fermi level. (**f**) The Zak phase from the occupied bands for both **a,b**. (**g**) The Zak phase from the occupied bands for **c–e**.

rise to a $\pm e/2$ surface polarization charge. The other route is through the surface states. In some NLSs, the drumhead surface state emerges. These two induce an appreciable dipole at the surface and may enhance surface Rashba splitting, as we show below.

This property is unique to materials with nodal lines. For a comparison, let us first consider an insulator with inversion and time-reversal symmetries. Consequently, the Zak phase satisfies $\theta(\mathbf{k}_\parallel) = 0$ or $\pi$ (mod $2\pi$) and it gives $\sigma(\mathbf{k}_\parallel) = Ne/2$ (where $N$ is an integer; see Supplementary Note 2). As $\theta(\mathbf{k}_\parallel)$ is continuous for all $\mathbf{k}_\parallel$, $N$ is common for all $\mathbf{k}_\parallel$. Thus far, no insulator with $N \neq 0$ is known, to our knowledge, possibly because of its instability due to the huge polarization charge at the surface. A non-zero even value of $N$ is not topologically protected and it is easily reduced to $N = 0$. Meanwhile, an odd value of $N$ means that dangling bonds, which are expected to be unstable, cover the surface. Thus, only the case $N = 0$ is physically realizable.

In materials with nodal lines, the Zak phase jumps by $\pi$ at the nodal lines; therefore, there are always two types of regions, one with $\theta(\mathbf{k}_\parallel) \equiv 0$ (mod $2\pi$) and one with $\theta(\mathbf{k}_\parallel) \equiv \pi$ (mod $2\pi$). The latter region inevitably leads to an appreciable polarization, as is exemplified by the Ca surface. In NLSs, the bulk charges eventually screen the polarization, but large deformation of the lattice structure and electronic relaxation (that is, screening) occur. In the present case, carriers screen the surface

polarizations, leaving behind dipoles at the surface. As roughly estimated for calcium (see Supplementary Note 5), the dipole density per surface unit cell is $\sim 5 \times 10^{-21}$ C·nm, the potential dip is $\sim -0.8$ eV at the surface and the electric field at the surface is $\sim 6.4$ V nm$^{-1}$.

Figure 5a shows the ratio of the interlayer distance at the surface to that of the bulk for the several surfaces of fcc Ca and Sr and the (001) surface of hexagonal close-packed (hcp) Be and Mg. The Ca and Sr surfaces, having nodal lines, are compressed near the surface by around 4% (equivalent to that in the bulk of Ca at $\sim 2$ GPa). This large compression in Ca and Sr may be associated with the surface charge induced by the nodal lines. From the above argument, the effect of the large charge imbalance at the surface is prominent only when the nodal lines are almost at the Fermi energy and no other Fermi surfaces exist. As shown in the Supplementary Note 5, carriers in the semimetal screen the surface charge through a screening length on the order of nanometres. Thus, the screening in this case is poor, because the NLS has a small number of carriers, leaving behind an appreciable dipole moment at the surface after the screening. On the other hand, if there are carriers other than those forming the nodal line, then the screening effect is much more prominent and dipoles at the surface are small. In Be and Mg, there are nodal lines[46,47] away from the Fermi level (at 0.0–1.1 eV in Be and at

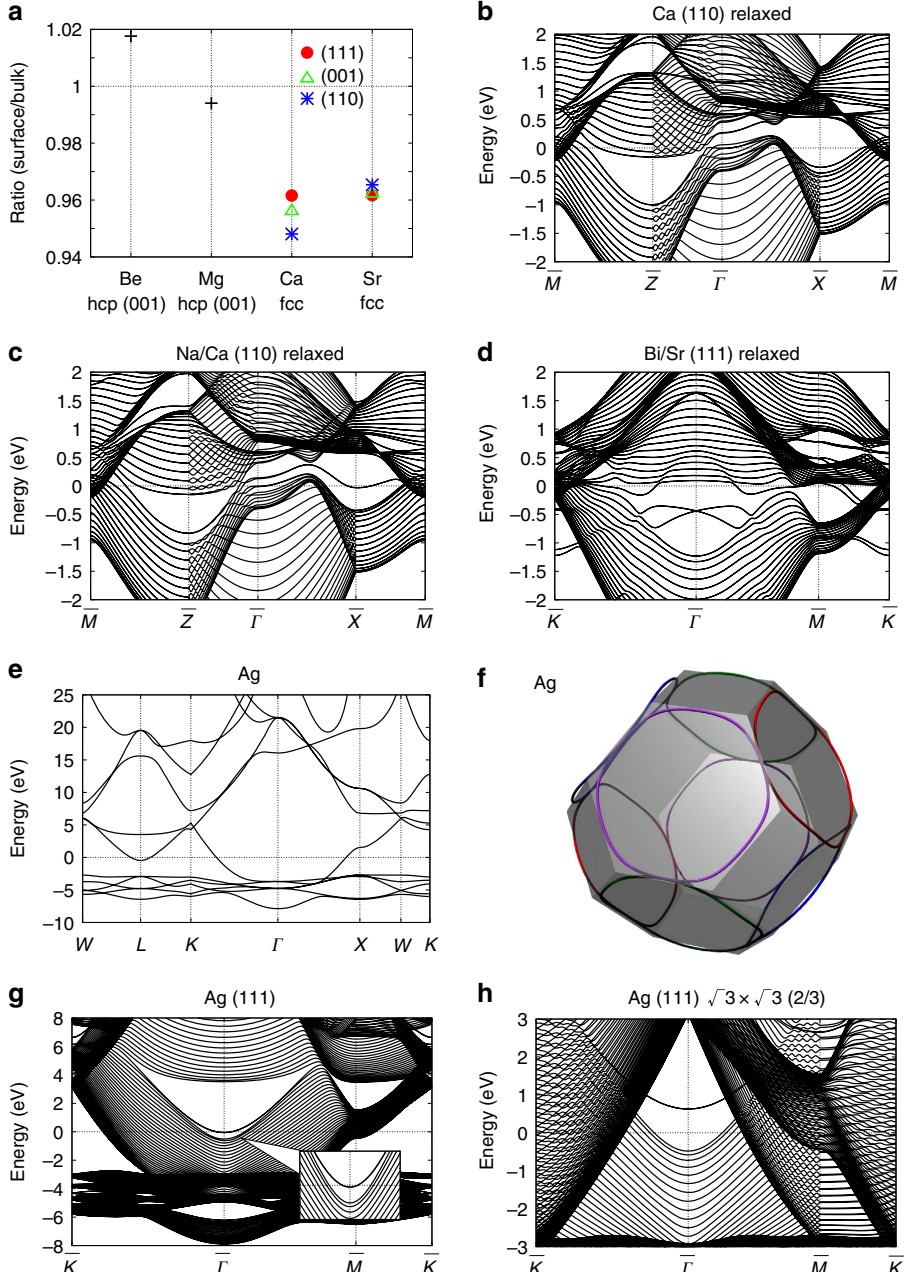

**Figure 5 | Nodal line and Rashba effect.** (**a**) Ratio of the distance between the layers at the surface to that of bulk (20 layers). (**b,c**) Electronic band structure of Ca for the (110) surface without Na and covered with Na, respectively, in the LDA. (**d**) Electronic band structure of Sr for the (111) surface covered with Bi in the LDA + SO. Structures in **a–d** are optimized by generalized gradient approximation (GGA). (**e**) Electronic band structure of Ag. (**f**) Nodal lines around 5 eV and the Brillouin zone of Ag, where the non-equivalent nodal line is shown in the different color. (**g,h**) Electronic band structure of Ag, without long-range $\sqrt{3} \times \sqrt{3}$ structure and with long-range $\sqrt{3} \times \sqrt{3}$ structure, respectively, on the (111) surface. The energy is measured from the Fermi level.

0.6–1.1 eV in Mg) and the density of states at the Fermi energy is large, because of the large Fermi surface; therefore, the compression of the lattice is small. In addition to the (111) surface of Ca, the (001) and (110) surfaces also show large deformations, because of large surface charges, where some **k** points with $0 \equiv 2\pi \pmod{2\pi}$ Zak phase are shown to have $\sigma(\mathbf{k}_\parallel) \equiv -e$ surface charge in our calculation (not shown). Thus, the large compression on the Ca surface is almost independent of the surface termination.

We find that the electronic relaxation alters the surface state dispersions (Fig. 5b) relative to the case with no relaxation (Fig. 3a). Meanwhile, the energies of some surface states are lowered by $\sim 1$ eV. This effect is attributed to the potential dip at

the surface due to surface charge, estimated as $\sim -0.8$ eV. This interpretation is also confirmed by subsequent calculation. When the instability originating from the surface charge is eliminated by covering the surface with alkali metal with low electronegativity, the surface states (Fig. 5c) emerge as almost the same as the original surface without relaxation (Fig. 3a).

We expect that such huge polarization charge due to the nodal lines would enhance Rashba spin splitting at the surface. In Fig. 5d, we show the Rashba splitting in a Bi monolayer on the Sr(111) surface. The Rashba splitting of the surface Bi 6$p$ bands near the Fermi level is as large as $E_R \sim 100$ meV. This large Rashba splitting may be partially attributed to the nodal lines; the potential dip at the surface due to the nodal lines gives rise to an

additional strong electric field within the Bi layer, roughly on the order of $1\,V/5\,\text{Å} \sim 2\,V\,nm^{-1}$. Such a strong additional electric field is expected to enhance the Rashba splitting. In addition to the nodal lines, the difference in the electronegativity between Sr and Bi may also enhance the Rashba splitting. Nonetheless, evaluating the contribution of the nodal line to the Rashba splitting is difficult, because the magnitude of the Rashba splitting is determined by the electric field very close to surface nuclei and by the asymmetry of the wavefunctions of the surface atoms[48]. For comparison we give some examples: the Rashba parameter increases by 0.005 and 0.011 nm·eV under an external electric field $E = 4.0\,V\,nm^{-1}$ on the Au(111)[49] and KTaO$_3$(001) surfaces[50], respectively. On the Bi/Sr(111) surface, the Rashba parameter for the Bi $6p_z$ band is 0.071 nm·eV, a part of which is attributed to the nodal lines.

For comparison, we discuss the Bi/Ag(111) surface, which is known to show large Rashba splitting[37]. We can also attribute this to nodal lines in Ag, but to a mechanism different from that in Bi/Sr(111). The conduction band structure of fcc Ag in Fig. 5e closely resembles that of Ca, Sr and Yb; Ag also has topological Dirac nodal lines around 5 eV (Fig. 5f). The Dirac nodal lines give rise to surface states for the (111) surface, which are visible around $\bar{\varGamma}$ near the Fermi energy, as shown in Fig. 5g. The well-known large Rashba splitting is realized when Bi atoms replace one-third of the surface Ag atoms, forming the $\sqrt{3} \times \sqrt{3}$ structure[37]. To establish the origin of the large Rashba splitting, in Fig. 5h we show the band structure for the $\sqrt{3} \times \sqrt{3}$ Ag(111) surface with one-third of the surface Ag atoms depleted (Fig. 1d); surface states also exist around 0.6 eV, which is higher than those in Fig. 5g because of depletion of some of the bonds. The Bi/Ag(111) surface is realized by adding Bi atoms to this surface and its surface states around the Fermi energy are formed by covalent bonding between Bi atoms and Ag surface states[51]. Therefore, the Ag surface states around 0.6 eV in Fig. 5h become stable by hybridization with the Bi $sp_z$ band and thus exhibit the large Rashba splitting.

Therefore, the nodal lines near the Fermi level enhance the Rashba splitting at the surface in two ways: one enhances via the surface charge, arising from the $\pi$ Zak phase, and the other is via hybridization with the emergent surface states from the nodal lines. The first scenario applies to Bi/Sr(111), whereas the second scenario occurs in Bi/Ag(111).

## Discussion

To summarize, *ab initio* calculation shows that the alkaline-earth metals Ca, Sr and Yb have topological nodal lines near the Fermi level in the absence of the SOI. Ca becomes an NLS at high pressure. The nodal lines do not lie on the mirror planes and have a solely topological origin characterized by the $\pi$ Berry phase. Consequently, the calculated Zak phase is either 0 or $\pi$, depending on the momentum. The $\pi$ Zak phase leads to a polarization charge at its surface region. After screening by carriers, the surface charge induces surface dipoles and a surface potential dip. The SOI gives rise to a small gap at the nodal line, whereas the surface states survive. Surface termination affects the surface states, whereas the Zak phase remains unaffected. In materials with nodal lines, both the large surface polarization charge and the emergent drumhead surface states can enhance Rashba splitting on the surface, as demonstrated in Bi/Sr(111) and in Bi/Ag(111).

## Methods

**Details of the first-principles calculation.** We calculate the band structures within the density functional theory. We use the *ab initio* code QMAS (Quantum MAterials Simulator) and OpenMX. The electronic structure is calculated using LDA with and without the relativistic effect (LDA and LDA + SOI, respectively). We also optimize the lattice parameter under pressure based on the generalized gradient approximation. The lattice parameter $a$ for Ca/Sr/Yb/Ag is 5.5884/6.0849/5.4847/4.0853 Å, respectively. The plane-wave energy cut-off is set to 40 Ry for Ca, Sr and Ag, and

50 Ry for Yb. The $12 \times 12 \times 12$ regular **k**-mesh including the $\varGamma$ point, with the Gaussian broadening of 0.025 eV, is employed. We construct an *spd* model for Ca, Sr, Yb and Ag from the Kohn–Sham bands, using the maximally localized Wannier function[52,53]. As the 4f orbital in Yb is fully occupied[54], we first construct the *spdf* model for Yb and disentangle out the 4f orbital from the model. We take 90 atoms on the (110) surface and 60 atoms on the (111) and (001) surfaces in the calculation of the electronic band structure (see Fig. 3b,d). Similarly, we use 120 atoms on the (001) surface with long-range $\sqrt{2} \times \sqrt{2}$ structure and 180 atoms on the (111) surface with long-range $\sqrt{3} \times \sqrt{3}$ structure (see Fig. 4c,e,f). We also use 30 atoms on the (110) surface with lattice relaxation and 20 atoms on the (111) surface with lattice relaxation (see Fig. 5b–d). Lastly, we use 45 atoms on the (111) surface without long-range $\sqrt{3} \times \sqrt{3}$ structure of Ag and 135 atoms with long-range $\sqrt{3} \times \sqrt{3}$ structure of Ag (see Fig. 5g,h). The electronic structure of Ag is calculated in the LDA (Fig. 5e,g,h). We take the vacuum region with thickness 2 Å for the slab calculation. Density of states at the Fermi level on the (110) surface is calculated by the surface Green's function[55,56] in which the system contains 720 atoms. (The electronic structure and lattice constant in Fig. 5 are calculated directly from the Kohn–Sham Hamiltonian and other results in Figs 2–5 are obtained via the Wannier function.)

**Conditions for emergence of nodal lines.** Mechanisms of the emergence of nodal lines involve either (A) mirror symmetry or (B) the $\pi$ Berry phase. For systems with mirror symmetry (A), the states on the mirror plane can be classified into two classes based on the mirror eigenvalues. If the valence and conduction bands have different mirror eigenvalues, then the two bands have no hybridization on the mirror plane, even if the two bands approach and cross. This results in a degeneracy along a loop on the mirror plane. This line-node degeneracy is protected by mirror symmetry; once the mirror symmetry is broken, the degeneracy is lifted in general. The second mechanism (B) occurs in spinless systems with inversion and time-reversal symmetries. The Berry phase around the nodal line is $\pi$. In spinless systems with inversion and time-reversal symmetries, the Berry phase along any closed loop is quantized as an integer multiple of $\pi$; therefore, the nodal line is topologically protected. Hence, closing of the gap occurs not at an isolated point in **k** space, but along a curve (that is, a nodal line) in general. If the band gap closes at some **k** by symmetry (such as twofold rotation), the closing of the gap is not limited to a single value of the wavevector **k**; instead, it extends along a nodal line in **k** space, as explained in detail in the Supplementary Note 1. In some materials the two mechanisms coexist, whereas in others the nodal line originates from only one of these mechanisms.

**Calculation of the Zak phase along a reciprocal vector G.** We separate the Bloch wavevector **k** into the components along $\mathbf{n} \equiv \mathbf{G}/|\mathbf{G}|$ and perpendicular to **n**: $\mathbf{k} = k_\perp \mathbf{n} + \mathbf{k}_\parallel$, $\mathbf{k}_\parallel \perp \mathbf{n}$. The integral with respect to $k_\perp$ is calculated with fixed $\mathbf{k}_\parallel$. For each value of $\mathbf{k}_\parallel$, one can define the Zak phase by

$$\theta(\mathbf{k}_\parallel) = -i \sum_n^{occ.} \int_0^{2\pi/a_\perp} dk_\perp \langle u_n(\mathbf{k}) | \nabla_{k_\perp} | u_n(\mathbf{k}) \rangle, \qquad (1)$$

where $u_n(\mathbf{k})$ is a bulk eigenstate in the $n$-th band, the sum is over the occupied states and $a_\perp$ is the size of the unit cell along the vector **n** (see Supplementary Note 2). The gauge is taken to be $u_n(\mathbf{k}) = u_n(\mathbf{k} + \mathbf{G})e^{i\mathbf{G}\cdot\mathbf{r}}$. The Zak phase is defined in terms of modulo $2\pi$. Under both inversion and time-reversal symmetries, the Zak phase is shown to take a quantized value 0 or $\pi$(mod $2\pi$)[12,45], as is also shown in the Supplementary Note 2. All of the cases presented in this paper satisfy the symmetry conditions for the Zak phase to be quantized as 0 or $\pi$. In one-dimensional insulators, the product of equation (1) and $e/(2\pi)$ is equal to the polarization, that is, the amount of the surface charge modulo $e$ (ref. 43). In three-dimensional insulators, the surface polarization charge $\sigma(\mathbf{k}_\parallel)$ at $\mathbf{k}_\parallel$ is calculated by multiplying the Zak phase by $e/(2\pi)$ and by integrating over the momentum $\mathbf{k}_\parallel$ perpendicular to **G**. Thus, this quantity is related to the surface polarization charge for the surface perpendicular to **n** (ref. 43).

**Data availability.** The authors declare that the data supporting the findings of this study are available within the paper and its Supplementary Information file.

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

## Acknowledgements

We thank Shoji Ishibashi for providing us with the *ab initio* code (QMAS) and pseudopotentials. This work was supported by the Grant-in-Aid for Scientific Research (numbers 26287062 and 26600012), by the Computational Materials Science Initiative (CMSI), Japan, and by the MEXT Elements Strategy Initiative to Form Core Research Center (TIES).

## Author contributions

All authors contributed to the main contents of this work. M.H. performed the *ab initio* calculation with contributions from T.M. R.O. constructed the arguments on the Zak phase and surface states. S.M. conceived and supervised the project. M.H., R.O. and S.M. drafted the manuscript. T.M. provided critical revisions of the manuscript.

## Additional information

**Competing financial interests:** The authors declare no competing financial interests.

**Publisher's note**: 

