## [Peer Review File · Nature Communications]

Reviewers' comments:

Reviewer #1 (Remarks to the Author):

Report on "Topological Dirac Nodal Lines in fcc Calcium, Strontium, and Ytterbium" by Motoaki Hirayama et al.

=====

This manuscript discusses an interesting observation, the presence of nodal lines in the bulk band structure of elemental alkaline earth metals. While the very existence of these nodal lines has been observed already 50 years ago (Refs. 34 and 35), the authors now try to establish a relation to modern notations in terms of Berry phase and topological origin.

I think that such a venture is timely and of interest for a broad community, since the subject are elemental metals which can be prepared in much simpler and better reproducible way than, e.g., ternary or quaternary compounds which are in the focus of most of the current research. However, I do not trust that the results are correct - see below. For this reason I suggest rejection of the manuscript but would encourage the authors to carefully check the situation in order to find out, if there is a way to remedy their arguments.

Major problems:

- The authors write that there are two reasons for the emergence of nodal lines, (A) mirror symmetry and (B) topology. They refer to the Methods section. There, both cases are explained in a few sentences. While I immediately understand the reasons for case (A), case (B) is not clear to me from the given explanation. Either, a reference should be given, or, a clear mathematical proof is needed for this topology-related matter.

- Most of the slab calculations are done in 3D supercell geometry using Wannier functions. (This is not very explicitly stated, but from all what I see and understand, it is the case.) The use of Wannier functions poses a serious accuracy problem for the position of the surface states. The authors briefly discuss the problem when comparing Fig. 3a with Fig. 5b.

However, they arrive at the (in my understanding wrong) conclusion that the difference is due to the neglect of surface lattice relaxation in the Wannier approach. What matters much more, are two other problems:

(i) The level shift due to the unavoidable surface dipole. No effect of electrostatic shifts due to the surface is included in the usual Wannier approach. Thus, the energy position of states close to the surface is not accurate. From the comparison of the two figures one finds that this shift amounts to about 0.5 eV in the case of Ca.

(ii) The basis may not be complete close to the surface, see e.g. Ref. [43].

Thus, no firm conclusion about the surface states position can be drawn from these calculations, including all related implications. Full self-consistent slab calculations would be needed.

- The calculation of Berry phases should be applied to the *bulk* system. The authors, however, seem to perform the calculation for 3D supercells simulating the surface. This means, that the results do not apply, e.g., for Ca but for an artificial crystal consisting of Ca stacks, separated by vacuum. This is my most important point.

Minor points:

- While most of the text is well understandable, there are a few spelling and grammar errors that

should be corrected.

- Fig. 2: "... nonequivalent nodal lines ..." To my mind, all lines shown in Fig. 2b are equivalent by cubic symmetry, as well as all lines shown in Fig. 2g.

- Pg. 2: "... is almost flat near the L points, ..." Well, the dispersion is about 0.3 eV, not so small.

- Pg. 2: "It gives four nodal lines ..." I count eight. The same problem is present at several places throughout the manuscript.

- Pg. 2: "... increases the energy of the p-orbital by about 1 eV ..." How is this number obtained?

- Fig. 3: "... in the surface region ..." should be quantified.

- Fig. 3: I would expect a different unit for a charge density.

- Fig. 3g and 3i: The state that crosses the Fermi level between K and Gamma (see inset to 3g) is clearly a surface state. This is obvious from Fig. 3i:

As soon as the state has crossed the Fermi level, charge is added to the two outermost layers on each side. I think that the problem is related with the (not quantified, see above) definition what a "surface state" is. Anyway, this observation makes a lengthy discussion on Pg. 3 invalid.

- Fig. 3g and 3h, discussion about the contribution of surface states to the polarization: The resolution is not very good, but I think that the jump of the Berry phase close to K (rightmost part of Fig. 3h) is related with the surface state crossing the Fermi level between M and K. Note: If you use 3D supercells, there is no fundamental difference between surface states and bulk states.

- Pg. 3: "... because the surface breaks inversion symmetry." I understand from other parts of your text, that you are using supercell with inversion center.

- Pg. 4: "We attribute this large splitting to the strong electric field from the nodal lines ..." This is something I would not undersign without a clear proof. The nodal lines may give a contribution (in k-space, but probably smeared in real space). This would have to be precisely quantified. Alternatively, did you check the case of Yb, were no nodal lines exist if spin-orbit coupling is considered?

- Methods: The k-integration method should be mentioned, not only the number of points.

- Methods: Which lattice parameter is used for the $p = 0$ calculations?

- Methods: Please give the size of the vacuum gap in the 3D supercells.

- Methods: Why is LDA used for the band calculations? Does GGA give any notable difference?

- Methods: The last remark is unclear. Fig. 5 shows nothing but electronic structure and lattice spacings. What has been calculated with Wannier functions in Fig. 5?

Reviewer #2 (Remarks to the Author):

In their manuscript titled "Topological Dirac Nodal Lines in fcc Calcium, Strontium, and Ytterbium", the

authors study a class of fcc alkaline earth metals, exploring the possibility of having topological nodal lines in their band structure. They also analyze how the surface of such systems is affected by the presence of nodal lines, proposing two mechanisms by which the Rashba effect could be enhanced. The field of topological properties of band insulators and semimetals is nowadays very active, and in this respect the manuscript is timely and could be potentially appealing for quite a broad audience. Nonetheless, in my opinion the manuscript in its present form is not strong enough to be published in Nature Communications. The title suggests that topological nodal lines can be found in fcc Ca, Sr and Yb, while actually such nodal lines are present only when spin-orbit interaction (SOI) is neglected. Furthermore, a topological robust nodal line is predicted only for Ca under pressure (again without SOI). The possible influence of nodal lines on surface Rashba effect seems to be more appealing, even though its importance and role in determining the strength of the Rashba splitting is in my opinion not fully supported by the present discussion. Overall, I found the manuscript not always clear and scientifically sound, as I try to elaborate in the following.

1) The authors differentiate between nodal lines emerging from topology or mirror symmetry. In my understanding, however, the origin of a nodal line always resides on the topological properties of the Bloch wave-functions in reciprocal space; as discussed in Ref [6], in systems without SOI the symmetry protection of nodal lines is provided by the composition T^*P of time-reversal (T) and parity (P) symmetries, whereas an additional mirror symmetry is required if a strong spin-orbit interaction is considered.

2) When discussing the nodal-line semimetal (NLS) properties of Ca, the authors say that the system can be classified as a weak NLS; then they say that Ca at ambient pressure is not a NLS, the latter phase being triggered by pressure. They discriminate between the two phases by looking (as far as I understand) at the curvature of the band dispersions around the L point. However, the classification proposed in Ref [6] is slightly different, and I would say more robust. First of all, the n -Berry phase (dubbed 1D Z2 invariant) around nodal lines is a signature of the topological protection of the T^*P symmetry; this guarantees the presence of nodal lines but not its protection against perturbations (the system can be gapped by adiabatically tuning $H(k)$). The nodal line is topologically stable only if the 3D system possesses a Z2 charge; quoting Ref. [6], "A nodal line with a Z2 charge can be considered a Z2 monopole, which can only be created or annihilated in pairs". In Ref. [6], a different 2D Z2 invariant has been proposed to verify if a nodal-line semimetal possesses a Z2 charge, classifying it as a topological (nodal-line) semimetal. Could the authors evaluate such 2D Z2 invariants in order to support their conclusions?

3) In agreement with Ref. [6], neither Sr nor Yb present nodal lines, due to their non-negligible SOI. I wonder if the results shown for Ca still hold when SOI is included.

4) I'm really confused by the discussion of bulk polarization. The formula of the Berry phase (Eq. (1) in Methods) is formally analogous to the polarization of one-dimensional systems; even in this case, however, the "real" polarization can be only defined by taking the difference of the Berry phase between two phases (the polar and the nonpolar ones) modulo a quantum. By no means the Berry phase of a single system can be related to its polarization; despite it can give informations about the charge distribution at the edges, I think that referring to it as a bulk polarization can be misleading. More technically, I do not fully understand how it is computed; is it calculated in slab geometry? Then the k_{\perp} is not defined. The authors should explicitly state if it has been calculated from the bulk model, in which case there is no breaking of inversion symmetry. On the other hand, if my understanding is correct, then it is not clear how such quantity has been evaluated in the case of depleted surfaces (Fig. 4 f) and g)).

5) There is some confusion between the π -Berry phase of loops around the nodal lines (that should be related to the 1D Z2 invariant of Ref. [6]) and the π -Berry phase of the "polarization", i.e., calculated along k_{\perp} . The authors should clarify their relationship before relating the π -Berry phase to surface states and/or surface charges, further clarifying which Berry-phase are referring to. For instance, at page 3 right column, they say "in ref. 6 it is noted that the Berry phase does not imply existence of surface states because the surface breaks inversion symmetry"; but the Berry-phase discussed in Ref. 6 is that calculated along loops enclosing the nodal line, not the one given in Eq (1) in Methods.

6) When discussing how the nodal lines affect the surface, the authors say that nodal line affects the "bulk polarization" (a claim that I do not understand, giving my previous comments) and may additionally cause the appearance of surface states. Could the authors better clarify the relationship between these two effects? What is the origin of the surface states induced by the presence of bulk nodal lines?

7) The most interesting section of the manuscript is the suggestion that nodal lines can affect Rashba effects of a Bi monolayer on the fcc alkali metals. However, while the suggestion that surface charge redistribution (evaluated via the Berry-phase "polarization") and/or the presence of surface states of the substrate may enhance the Rashba effect of the Bi is indeed interesting, the informations provided in the manuscript are in my opinion not enough to support such claim. For instance, how large is the Rashba splitting of Bi monolayers on substrates without nodal lines? Can the authors estimate the enhancement of the effect with and without nodal lines? Furthermore, the authors claim that Ag substrate contribute to Rashba splitting mostly via hybridization of Bi with emergent surface states arising from nodal lines; did they calculate the surface charge redistribution (using the Berry-phase "polarization"), in order to rule out its role?

Reviewers' comments:

Reviewer #1 (Remarks to the Author):

The authors have provided explicit response to all questions raised, and many of the issues from both reviewers are appropriately addressed.

Yet there are, in my view, open questions:

- Topological origin of the nodal lines (extending my first comment of the first review):

The authors find, that the Zak phase around the nodal lines is π and conclude that this points to a topological origin. I am not sure, if this is justified. Could the nodal lines be due to a symmetry different from the mirror symmetry? The authors introduce the notation "weak" NLS, but before accepting a new notation, one has to be sure about its meaning.

Here, it seems to me that the causality is not clear: The nodal lines are there, and only after their identification the Zak phase can be calculated. So, is there any primary topological reason for their existence? If such a reason cannot be identified, one cannot talk about "topological origin".

- In most of the text, the authors say that the surface polarization can be obtained by evaluating the area in k -parallel, where the Zak phase is not zero. Suddenly, on Pg. 5, this is abrogated by telling that also the (001) and (110) surfaces show a large polarization due to jumps of the

phase by 2π . This means, there is no clear prediction possible about the nodal line impact on Rashba splitting: It is only one of several additive contributions, and not necessarily the dominating one.

Technical:

- The grammar would need to be corrected by a native speaker.

- Parts of the discussion are hard to follow, not always with a clear logic, and with jumps in the argumentation.

- Ref 14 is incomplete; Ref. 2 of the Supplement is wrong.

- Supplement, "wavefunction" should read "vector" before Eq. (8).

Reviewer #2 (Remarks to the Author):

The authors have significantly improved their manuscript, which I think might be suitable for publication in Nature Communications once the following minor remarks have been addressed.

The main result of the manuscript is in my opinion the relationship between the bulk nodal lines and the surface charges (rather than states), which could possibly have consequences on the surface Rashba splitting; the title, however, seems to me a bit out of focus, referring to topological nodal lines in elemental metals (which, by the way, are not so robust with respect to spin-orbit interaction, as also stated in the main text; on the other hand, the authors suggest that the bulk/surface correspondence should survive for moderate SOI even in the presence of small SOI-induced gap, implying that the latter, rather than the presence of topological nodal lines, is the meaningful result).

While I think that the manuscript readability has been significantly improved with respect to the definition of Berry phase and Zak phase, in order to avoid further misunderstandings I would suggest the authors to relate the Zak phase to a surface charge, rather than polarization, surface polarization or topological polarization (as in fact done in Ref 43)

As it stands, the conclusive paragraph of the section titled "Zak phase and surface states", where surface depletion is considered, seems unrelated to the previous discussion, being unclear what conclusion can be drawn from the numerical results. Is it the robustness of Zak phase/ surface charges? I would suggest the authors to add a clear conclusive sentence at the end of the paragraph also and not only in the summary section.

Eventually, while the proposed mechanism (surface charge and reduced screening due to the presence of nodal lines, leading to an increase of the surface electric field) for enhanced surface Rashba splitting is reasonable, there is no clear evidence of its prominence, since several other factors are at play which can be hardly ruled out (as admitted by the authors themselves). Even though I believe it is an interesting proposal, I would suggest the authors to smooth their claims about the enhancement of Rashba splitting as due to the presence of nodal lines near the Fermi level, clarifying that it is, in fact, a speculation.

REVIEWERS' COMMENTS:

Reviewer #1 (Remarks to the Author):

Most remarks of my previous report were considered by the authors and related necessary changes to the manuscript were implemented. There is one technical point: Ref. [S1] (previous Ref. [S2]) is still wrong.

Reviewer #2 (Remarks to the Author):

I believe that the authors have replied to all comments and issues previously raised in a satisfactory way. I have no further comments and I think that the manuscript is suitable for publication in Nature Communications

Reply to First Report of Reviewer #1

This manuscript discusses an interesting observation, the presence of nodal lines in the bulk band structure of elemental alkaline earth metals. While the very existence of these nodal lines has been observed already 50 years ago (Refs. 34 and 35), the authors now try to establish a relation to modern notations in terms of Berry phase and topological origin.

I think that such a venture is timely and of interest for a broad community, since the subject are elemental metals which can be prepared in much simpler and better reproducible way than, e.g., ternary or quaternary compounds which are in the focus of most of the current research. However, I do not trust that the results are correct - see below. For this reason I suggest rejection of the manuscript but would encourage the authors to carefully check the situation in order to find out, if there is a way to remedy their arguments.

Reply: Thank you very much for the helpful comments. We revised the manuscript in the light of the comments as described below. We believe that the revised manuscript warrants publication in Nature Communications.

Comment #1: The authors write that there are two reasons for the emergence of nodal lines, (A) mirror symmetry and (B) topology. They refer to the Methods section. There, both cases are explained in a few sentences. While I immediately understand the reasons for case (A), case (B) is not clear to me from the given explanation. Either, a reference should be given, or, a clear mathematical proof is needed for this topology-related matter.

Reply to Comment #1: The mechanism (B) is characterized by the π Berry phase. Namely, the Berry phase around the nodal line is equal to π . In spinless systems with inversion and time-reversal symmetries, the Berry phase along any closed loop is quantized as an integer multiple of π , and therefore, the nodal line cannot disappear by a small perturbation. This is a topological protection of the nodal line in the mechanism (B). In the previous manuscript we called (B) as “topology” mechanism, but this terminology might be ambiguous in the light of the Reviewer’s Comment. So in the revised manuscript, we call this mechanism as “ π Berry phase”, and added some explanations on this mechanism in the Methods Section and in the Supplementary

Material.

Comment #2: *Most of the slab calculations are done in 3D supercell geometry using Wannier functions. (This is not very explicitly stated, but from all what I see and understand, it is the case.) The use of Wannier functions poses a serious accuracy problem for the position of the surface states. The authors briefly discuss the problem when comparing Fig. 3a with Fig. 5b.*

However, they arrive at the (in my understanding wrong) conclusion that the difference is due to the neglect of surface lattice relaxation in the Wannier approach. What matters much more, are two other problems:

(i) The level shift due to the unavoidable surface dipole. No effect of electrostatic shifts due to the surface is included in the usual Wannier approach. Thus, the energy position of states close to the surface is not accurate. From the comparison of the two figures one finds that this shift amounts to about 0.5 eV in the case of Ca.

(ii) The basis may not be complete close to the surface, see e.g. Ref. [43].

Thus, no firm conclusion about the surface states position can be drawn from these calculations, including all related implications. Full self-consistent slab calculations would be needed.

Reply to Comment#2

In our paper, some of the slab calculations are done in 3D supercell geometry using Wannier functions, while others are not. As the Reviewer points out in (i) and (ii), the calculation of the electronic structure using the Wannier function is inaccurate for the surface. Especially, the screening effect against the topological polarization would be dominant. However, this point does not change our argument except for minor modifications. We rather think the result using the Wannier function is necessary for the understanding of physics of nodal lines by comparing with that in the DFT with relaxation. Actually, we show the topological nature of fcc alkali earth metals in the following two-step approach.

First, we calculate the surface states from the Wannier function obtained from the bulk Hamiltonian. Because the topological surface state originates from the topology of the bulk, it serves as a starting point for the discussion to calculate what surface state is required purely from the bulk Hamiltonian. If we skip this analysis using the Wannier function, it is difficult to separate the topological effect from the effect of surface reconstruction.

Next, we consider the effects onto the surface states both from the lattice and from electronic relaxations by the DFT/GGA (Figs. 5a-d). The reason for this second step is that the calculation based on the Wannier function is inaccurate for the surface state, because of the relaxation and of the insufficiency of the basis functions, as the Reviewer #1 pointed out in (i) and (ii).

Through this two-step approach, we conclude that the relaxation due to the huge topological polarization is unique for the fcc alkaline earth metals, since the nodal-line exists at the Fermi level.

The surface polarization charge will be screened by bulk carriers and produce surface dipoles. To elucidate the effect of surface dipole, we add a simple model calculation on dipoles induced by the screening by using the Poisson equation, as explained in the Supplemental Material. A rough estimate of the potential dip due to the surface dipole is about -0.77eV , which is in good agreement with the comparison between Fig. 3a and Fig. 5b. The key factor for the formation of the surface dipole is the bulk carrier density; the small carrier density in Ca leads to poor screening by the carriers, and is helpful for formation of surface dipoles.

We also studied the effect of lattice relaxation on the electronic structure and it turns out to be small. Here, we show the electronic band structure of Ca (110) surface with/without lattice relaxation in Figs. R1 a, where the black solid line is same as FIG.5 in the manuscript, and b, where the distance between the layers at the surface is changed from that of the bulk. Although the lattice constant is largely different from each other at the surface, the electronic band structure near the Fermi level is almost same. Thus it turns out that the lattice relaxation is not so important in the present case, while the electronic relaxation (i.e. screening) plays a major role in the surface state dispersions.

We thank the Reviewer to remind us the importance of the electronic relaxation. We have modified the manuscript accordingly as below.

(OLD) The lattice relaxation alters the surface state dispersions (Fig. 5b) as compared from the case with no relaxation (Fig. 3a). When the relaxation is eliminated by covering the surface with alkali metal with small electronegativity, the surface states, shown in Fig. 5c, turn out to be almost the same as the original surface without

relaxation (Fig. 3a).

(NEW) The electronic relaxation alters the surface state dispersions (Fig. 5b) as compared from the case with no relaxation (Fig. 3a). When the instability originating from the topological polarization is eliminated by covering the surface with alkali metal with small electronegativity, the surface states, shown in Fig. 5c, turn out to be almost the same as the original surface without relaxation (Fig. 3a).

FIG. R1. Electronic band structure of Ca for the (110) surface in the DFT/LDA. a, that with/without lattice relaxation (black dotted line/red solid line). b, the distance between the layers at the surface is changed from that of the bulk.

Comment #3: *The calculation of Berry phases should be applied to the *bulk* system. The authors, however, seem to perform the calculation for 3D supercells simulating the surface. This means, that the results do not apply, e.g., for Ca but for an artificial crystal consisting of Ca stacks, separated by vacuum.*

This is my most important point.

Reply to Comment #3: We calculated Zak phase (Berry phase) not for slab but for bulk wavefunction. Figure 3h shows the Zak phase along the (111) direction calculated from the bulk wavefunctions. As shown in the paper by Vanderbilt and King-Smith (PRB48,4442 (1993)), this bulk quantity is related with the polarization charge at the surface. Namely, we use this bulk-surface correspondence to discuss surface polarization charge in terms of BULK wavefunctions. To avoid misunderstanding, we

added explanations on the calculation of the Berry phase in the text and in the Methods section.

Comment #4: While most of the text is well understandable, there are a few spelling and grammar errors that should be corrected.

Reply to Comments #4: We thoroughly have checked the text and corrected the errors.

Comment #5: Fig. 2: "... nonequivalent nodal lines ..." To my mind, all lines shown in Fig. 2b are equivalent by cubic symmetry, as well as all lines shown in Fig. 2g.

Comment #6: Pg. 2: "It gives four nodal lines ..." I count eight. The same problem is present at several places throughout the manuscript.

Reply to Comments #5 and #6: In Ca and Sr, there are four nodal lines, and they are related by symmetry operations of the cubic symmetry. In Fig.2b, there seem to be eight nodal lines, but two nodal lines at the opposite faces of the Brillouin zone are in fact equivalent.

We use the word "equivalent" in the sense that they are identical (up to modulo reciprocal lattice vectors), whereas the Reviewer #1 interpreted the word "equivalent" as "related by crystallographic symmetry". To avoid misunderstanding we have rewritten the text by removing the word "equivalent".

Comment #7: Pg. 2: "... is almost flat near the L points, ..." Well, the dispersion is about 0.3 eV, not so small.

Reply to Comment #7: Here, we discuss the relative scale of the dispersion of the upper band (0.3eV) at the L points against the lower one (1.8eV). To clarify the meaning we revised the text from "almost flat" to "relatively flat".

Comment #8: Pg. 2: "... increases the energy of the p-orbital by about 1 eV ..." How is this number obtained?

Reply to Comment #8: We calculate the maximally localized Wannier function of the *spd*-orbitals from the DFT/LDA band structure near the Fermi level, and check the on-site potential of the Wannier orbital both at ambient and under pressure.

Anyway, we deleted the description of the value "1eV" from the text, because this

number is not important in the subsequent argument, and it is not straightforwardly seen in the Figures in the paper.

Comment #9: *Fig. 3: "... in the surface region ..." should be quantified.*

Reply to Comments #9: The surface region here refers to one atomic layer near the surface, and we added this explanation in the caption of Fig.3.

Comment #10: *I would expect a different unit for a charge density.*

Reply to Comment #10: In Fig.3i, the vertical axis represents the charge density per 2D unit cell (along the surface) within each atomic layer, measured in the unit of the electronic charge e . We added the explanation on it.

Comment #11: *Fig. 3g and 3i: The state that crosses the Fermi level between K and Gamma (see inset to 3g) is clearly a surface state. This is obvious from Fig. 3i:*

As soon as the state has crossed the Fermi level, charge is added to the two outermost layers on each side. I think that the problem is related with the (not quantified, see above) definition what a "surface state" is. Anyway, this observation makes a lengthy discussion on Pg. 3 invalid.

Reply to Comment #11: First of all, the state that crosses the Fermi level between K and Gamma (see inset to 3g) is NOT a surface state, and it is an important point in the paper.

Below we explain the reason for this statement.

Surface states are defined as states with finite penetration depth. It is defined when the system size is infinite. In this sense, the state that crosses the Fermi level between K and Gamma is not a surface state, because as the system size becomes larger, the gap between the bulk valence band and the bulk conduction band converges to zero and the state that crosses the Fermi level between K and Gamma disappears. Therefore this state is not a surface state. The small gap in Fig.3g is a minigap due to the finite-size effect, and it vanishes in the infinite-size system.

Instead, Fig.3i shows that this deficiency of charge at the surface is due to bulk states. It is fully consistent with our calculation results of the bulk Berry phase, i.e. the bulk

charge polarization shows $e/2$ at $\forall\text{Lambda}_1$ and 0 at $\forall\text{Lambda}_2$.

Furthermore, as explained in the paper, any charge imbalance originated from the surface state, if any, should be an integer multiple of the electron charge e per surface unit cell, and so it cannot explain the charge deficiency of $e/2$ observed in Fig.3i.

We added these detailed explanations on this issue in the text.

Comment #12: Fig. 3g and 3h, discussion about the contribution of surface states to the polarization: The resolution is not very good, but I think that the jump of the Berry phase close to K (rightmost part of Fig. 3h) is related with the surface state crossing the Fermi level between M and K. Note: If you use 3D supercells, there is no fundamental difference between surface states and bulk states.

Reply to Comment #12: As stated above, in the infinite-size limit there is fundamental difference between surface states and bulk states, and only in this limit one can relate the bulk Berry phase with the surface charge polarization. Otherwise there is no clear meaning in discriminating surface states from bulk states.

To show more clearly the dependence of the Berry phase on the wavevector and to improve resolution of Fig.3h, we replaced Fig. 3h to the figure showing the whole surface Brillouin zone (from the previous one where only the wavevectors along high-symmetry lines are shown).

Again, as we explained in the Reply to Comment #11, the jump of the Berry phase cannot be explained from surface states from abovementioned multiple reasons.

Comment #13: Pg. 3: "... because the surface breaks inversion symmetry." I understand from other parts of your text, that you are using supercell with inversion center.

Reply to Comment #13: This argument of the relationship between the Berry phase and the surface states is unrelated with the supercells. Anyway, the statement in the previous manuscript was misleading, and so we removed the phrase "because the surface breaks inversion symmetry" from the text.

(The argument in the paper by Fang et al. (ref.43) is the following: the topological character coming from the $\forall\pi$ Berry phase is coming from inversion and time-reversal symmetries, and so in the presence of a surface, inversion symmetry is lost and this topological character becomes vague. Therefore, it cannot conclude existence of surface

states.)

Comment #14: *Pg. 4: "We attribute this large splitting to the strong electric field from the nodal lines ..." This is something I would not undersign without a clear proof. The nodal lines may give a contribution (in k-space, but probably smeared in real space). This would have to be precisely quantified. Alternatively, did you check the case of Yb, were no nodal lines exist if spin-orbit coupling is considered?*

Reply to Comment #14:

Yes, we checked the case of Yb in several points. Even if the SOI is considered, the effect of the nodal line without the SOI mostly survives by continuity. For example the drumhead surface states remain (see Fig. 3d).

Furthermore, the polarization $\forall \sigma(k//)$ far from the nodal line is expected to take a similar value as in the cases without SOI, because the effect of the SOI is prominent only near the nodal lines.

Nevertheless, the Rashba SOC is expected to be smaller in Yb, because the size of the nodal line around the W point in Yb is very small.

Comment #15: *Methods: The k-integration method should be mentioned, not only the number of points.*

Reply to Comment #15:

The 12x12x12 regular k-mesh including the Gamma point, with the Gaussian broadening of 0.025 eV, is employed. We have revised the Methods section accordingly.

Comment #16: *Methods: Which lattice parameter is used for the p = 0 calculations?*

Comment #17: *Methods: Please give the size of the vacuum gap in the 3D supercells.*

Reply to Comment #16 and #17: The lattice parameter a for Ca/Sr/Yb/Ag is 5.5884/6.0849/5.4847/4.0853 Å. We take 20Å as the size of the vacuum region for the slab calculation. We added this information to the Methods Section.

Comment #18: *Methods: Why is LDA used for the band calculations? Does GGA give any notable difference?*

Reply to Comment #18: We have used the LDA+SO simply because the GGA is not

implemented in the fully-relativistic version of the QMAS code. We expect that the GGA+SO gives similar results as the LDA+SO, since experiences for non-spin-orbit calculations tell us that the gradient correction has minor impact on the electronic structure.

Comment #19: *Methods: The last remark is unclear. Fig. 5 shows nothing but electronic structure and lattice spacings. What has been calculated with Wannier functions in Fig. 5?*

Reply to Comment #19: Fig.5f is calculated from the bulk Wannier function. The other results in Fig.5 are directly obtained from the DFT/LDA. In particular, Fig.5 a-d contain the effect of the lattice relaxation.

Reply to First Report of Reviewer #2

Overall Comment: *In their manuscript titled "Topological Dirac Nodal Lines in fcc Calcium, Strontium, and Ytterbium", the authors study a class of fcc alkaline earth metals, exploring the possibility of having topological nodal lines in their band structure. They also analyze how the surface of such systems is affected by the presence of nodal lines, proposing two mechanisms by which the Rashba effect could be enhanced. The field of topological properties of band insulators and semimetals is nowadays very active, and in this respect the manuscript is timely and could be potentially appealing for quite a broad audience. Nonetheless, in my opinion the manuscript in its present form is not strong enough to be published in Nature Communications. The title suggests that topological nodal lines can be found in fcc Ca, Sr and Yb, while actually such nodal lines are present only when spin-orbit interaction (SOI) is neglected. Furthermore, a topological robust nodal line is predicted only for Ca under pressure (again without SOI). The possible influence of nodal lines on surface Rashba effect seems to be more appealing, even though its importance and role in determining the strength of the Rashba splitting is in my opinion not fully supported by the present discussion. Overall, I found the manuscript not always clear and scientifically sound, as I try to elaborate in the following.*

Reply: Thank you very much for the helpful comments. We revised the manuscript in the light of the comments as described below. We believe that the revised manuscript

warrants publication in Nature Communications.

Comment 1): *The authors differentiate between nodal lines emerging from topology or mirror symmetry. In my understanding, however, the origin of a nodal line always resides on the topological properties of the Bloch wave-functions in reciprocal space; as discussed in Ref [6], in systems without SOI the symmetry protection of nodal lines is provided by the composition T^*P of time-reversal (T) and parity (P) symmetries, whereas an additional mirror symmetry is required if a strong spin-orbit interaction is considered.*

Reply to Comment 1): In the cases with and without SOI, mirror symmetry alone can guarantee existence of nodal lines, and it is called mechanism (A) in our paper. It is explained in Methods Section.

As discussed in Ref.[6], additional time-reversal (T) and parity (P) symmetries will give topological characteristics to the nodal lines from the mechanism (A). Nevertheless, even without T or P symmetries, nodal lines can appear solely from mirror symmetry. So it is appropriate to call this (A) as based on mirror symmetry.

Anyway, as discussed in the Reply to Reviewer #1's Comment #1, we revised the paper to call the mechanism (B) as “ π Berry phase” to avoid ambiguity and misunderstanding.

Comment 2) *When discussing the nodal-line semimetal (NLS) properties of Ca, the authors say that the system can be classified as a weak NLS; then they say that Ca at ambient pressure is not a NLS, the latter phase being triggered by pressure. They discriminate between the two phases by looking (as far as I understand) at the curvature of the band dispersions around the L point. However, the classification proposed in Ref [6] is slightly different, and I would say more robust. First of all, the Berry phase (dubbed 1D Z2 invariant) around nodal lines is a signature of the topological protection of the T^*P symmetry; this guarantees the presence of nodal lines but not its protection against perturbations (the system can be gapped by adiabatically tuning $H(k)$). The nodal line is topologically stable only if the 3D system possesses a Z2 charge; quoting Ref. [6], "A nodal line with a Z2 charge can be considered a Z2 monopole, which can only be created or annihilated in pairs". In Ref. [6], a different 2D Z2 invariant has been proposed to verify if a nodal-line semimetal possesses a Z2 charge,*

classifying it as a topological (nodal-line) semimetal. Could the authors evaluate such 2D Z2 invariants in order to support their conclusions?

Reply to Comment 2): Here the topological invariant discussed in this paper is different from the Z2 topological number discussed in Ref. [6]. In the present paper, the topological invariant is the $\forall \pi$ Berry phase around the nodal line. This topologically protects the nodal line as long as the nodal line has a finite size in k space, while it can disappear after shrinking to a point. Meanwhile in Ref. [6] the authors discuss the Z2 topological number; if it is nontrivial, it will prohibit the nodal line to disappear after shrinking into a point. In this sense the Z2 topological number in Ref.[6] protects the nodal line in a stronger manner.

In the present cases of alkaline earth metals, the Z2 topological number (defined in Ref.[6]) is trivial, because the nodal lines can disappear by shrinking into a point, when we add on-site energy onto 4s orbitals by hand.

In the text we added comments on the Z2 topological number defined in Ref.[6], and explain that our case is trivial in the sense of the Z2 topological number of Ref.[6], with its details given in the Supplementary Material.

Comment 3) In agreement with Ref. [6], neither Sr nor Yb present nodal lines, due to their non-negligible SOI. I wonder if the results shown for Ca still hold when SOI is included.

Reply to Comment 3): The nodal lines are gapped if the SOI is included. Nevertheless, even if the SOI is considered, the effect of the nodal line without the SOI mostly survives by continuity. For example the drumhead surface states remain (see Fig. 3d). Furthermore, the polarization $\forall \sigma(k//)$ far from the nodal line is expected to take a similar value as in the cases without SOI, because the effect of the SOI is prominent only near the nodal lines.

Nevertheless, the Rashba SOC is expected to be smaller in Yb, because the size of the nodal line around the W point in Yb is very small.

Comment 4) I'm really confused by the discussion of bulk polarization. The formula of the Berry phase (Eq. (1) in Methods) is formally analogous to the polarization of one-dimensional systems; even in this case, however, the "real" polarization can be only defined by taking the difference of the Berry phase between to phases (the polar and the

nonpolar ones) modulo a quantum. By no means the Berry phase of a single system can be related to its polarization; despite it can give informations about the charge distribution at the edges, I think that referring to it as a bulk polarization can be misleading. More technically, I do not fully understand how it is computed; is it calculated in slab geometry? Then the k_{\perp} is not defined. The authors should explicitly state if it has been calculated from the bulk model, in which case there is no breaking of inversion symmetry. On the other hand, if my understanding is correct, then it is not clear how such quantity has been evaluated in the case of depleted surfaces (Fig. 4 f) and g)).

Reply to Comment 4): In the paper by Vanderbilt and King-Smith (PRB48,4442 (1993) , ref.43), the authors show that the surface polarization charge is indeed given by the integral of the Berry phase over the surface Brillouin zone modulo e/A_{surface} , where A_{surface} is the size of the surface unit cell. Thus, while the difference of polarization is indeed given by the difference of the Berry phase between two phases (as the Reviewer #1 points out), the polarization **modulo e/A_{surface}** is indeed given by the Berry phase. In the present paper, we care only the value the polarization modulo e/A_{surface} and so the use of the Berry phase suffices.

The problem of depleted surfaces is related with the choice of unit cell. In reality, for the given surface, there are various ways to choose the unit cell. While this choice affects the electronic polarization, the total polarization which is the sum of the electronic and ionic ones is unaffected as shown in Ref. [43]. Therefore for convenience in our paper, we choose the unit cell such that the ionic polarization is zero. By this choice, the depleted surfaces can be treated as well.

We admit that the presentation of the previous version of our paper was not so clear in the relationship between surface charge polarization and the Berry phase. Therefore we added some explanations and discussions. In fact in the previous version of our paper such explanations were omitted because it is mostly a known result in ref. [43] (PRB48, 4442 (1993)); but to improve presentation we added these explanations. In particular, we improved the presentation so that the readers can easily see that the Zak phase is a bulk quantity.

Comment 5) *There is some confusion between the Berry phase of loops around the nodal lines (that should be related to the 1D Z2 invariant of Ref. [6]) and the Berry phase of*

the "polarization", i.e., calculated along k_{\perp} . The authors should clarify their relationship before relating the Berry phase to surface states and/or surface charges, further clarifying which Berry-phase are referring to. For instance, at page 3 right column, they say "in ref. 6 it is noted that the Berry phase does not imply existence of surface states because the surface breaks inversion symmetry"; but the Berry-phase discussed in Ref. 6 is that calculated along loops enclosing the nodal line, not the one given in Eq (1) in Methods.

Reply to Comment 5): Because the Berry phase around the nodal line is π , the Berry phase (Zak phase), i.e. an integral of the Berry connection along the fixed k_{\perp} has a jump by π . To avoid confusion, we call $\theta(k_{\perp})$ as a Zak phase, which is a special name of the Berry phase along a certain reciprocal vector G .

When k_{\perp} is changed across the projection of the nodal line along the \mathbf{b}_n direction. We added an explanation on this point in the Supplementary Material with a figure.

Comment 6) When discussing how the nodal lines affect the surface, the authors say that nodal line affects the "bulk polarization" (a claim that I do not understand, giving my previous comments) and may additionally cause the appearance of surface states. Could the authors better clarify the relationship between these two effects? What is the origin of the surface states induced by the presence of bulk nodal lines?

Reply to Comment 6): It is indeed true that within the effective model calculation, the nodal line leads to existence of "drumhead" surface states, as shown in the paper by Kim, Wieder, Kane and Rappe (PRL 115, 03680 (2015), ref. 9). While drumhead surface states exist in many materials and systems with nodal lines, it is not always the case, as is explained previously and in our paper. In fact in the effective model calculation, one type of a boundary condition is assumed, whereas the surface states do crucially depend on boundary conditions, and sometimes do not exist.

(We note that only in systems with chiral symmetries, existence of the nodal lines leads to existence of drumhead surface states, as discussed in our paper.)

Comment 7) The most interesting section of the manuscript is the suggestion that nodal lines can affect Rashba effects of a Bi monolayer on the fcc alkali metals. However, while the suggestion that surface charge redistribution (evaluated via the Berry-phase

"polarization") and/or the presence of surface states of the substrate may enhance the Rashba effect of the Bi is indeed interesting, the informations provided in the manuscript are in my opinion not enough to support such claim. For instance, how large is the Rashba splitting of Bi monolayers on substrates without nodal lines? Can the authors estimate the enhancement of the effect with and without nodal lines? Furthermore, the authors claim that Ag substrate contribute to Rashba splitting mostly via hybridization of Bi with emergent surface states arising from nodal lines; did they calculate the surface charge redistribuiton (using the Berry-phase "polarization"), in order to rule out its role?

Reply to Comment 7): What we can show from the nodal line is the existence of large dipole induced at the surface due to the polarization coming from π Berry phase.

First, the induced surface charge is calculated by the bulk Berry phase polarization, and carriers in the semimetal screen the surface charge, resulting in surface dipoles and potential dip near the surface. We calculated them with the help of Poisson equation. The surface dipole density is roughly estimated as $4.7 \times 10^{-21} \text{C} \cdot \text{nm}$ per surface unit cell, with the potential depth at the surface is -0.77eV . The electric field is estimated as 6.4V/nm . Details are added in Supplementary Material.

Because the Berry phase is a bulk quantity, the lattice relaxation and screening by the surface charge cannot be calculated from the Berry phase alone. Our conclusion is that there are large dipole moments at the surface due to nodal lines, which may appreciably enhance the Rashba SOI.

> how large is the Rashba splitting of Bi monolayers on substrates without nodal lines?

The (freestanding) Bi monolayer has no Rashba SOI because it is inversion symmetric. Only via a hybridization with the substrate, the inversion symmetry is broken and Rashba SOI appears. The surface dipole is therefore favorable for inducing a large SOI at the surface.

> Can the authors estimate the enhancement of the effect with and without nodal lines?

It is not possible to estimate the enhancement of the effect with and without nodal lines, because the Rashba SOI is affected by various factors, such as lattice relaxation, bulk band structure, and field distribution very close to surface nuclei (cf. Bihlmayer et al., Surf. Sci 600, 3888 (2006)), apart from nodal lines. Even when we modify the system by hand in some way to remove the nodal line for the purpose of evaluating the contribution of nodal lines, it will affect such multiple factors affecting the Rashba SOI.

Hence, we cannot single out the effect of nodal line.

Here what we can do is to physically discuss the effect of nodal lines. In the Supplementary Material we added the section “screening in nodal-line semimetals” to show that the smallness of carrier concentration in nodal-line semimetals is crucial in making the screening to be poor, leaving behind a large dipoles at the surface.

Furthermore, to see the effect of nodal-lines we added some explanations on Be and Mg, which are isovalent to Ca and Sr but have different lattice structure, with a large Fermi surface. In these materials, since the nodal lines are deep below the Fermi energy and they have large carrier concentration, the surface charging effect is expected to be much smaller, and it is indeed the case. From this comparison, we can guess that existence of nodal lines around the Fermi energy and smallness of the carrier concentration are crucial in giving a large Rashba SOI.

=====
=====
=====
=====

List of corrections

1) In the two typical mechanisms of the nodal lines, we change the terminology from “(B) topology” to “(B) π Berry phase”, and added some explanations on this mechanism in the Methods Section and in the Supplementary Material.

2) In the previous manuscript, the Berry phase around the nodal line and the Berry phase along a reciprocal vector are both called Berry phase. For clarity of presentation, in the revised manuscript we call the latter as “Zak phase”. We added an explanation on the relationship between these two phases in the main text and in the Supplemental Material. Namely, when k_{\perp} is changed across the projection of the nodal line along the \mathbf{m}_n direction.

3) we added a note on the number of nodal lines, i.e. there are 4 nodal lines in Ca and Sr, which are mutually related by symmetry. In addition, to avoid misunderstanding we have rewritten the text by removing the word “equivalent”.

4) In the description of the bands near the L points, we revised the text from “almost flat”

to “relatively flat”.

5) In the revised manuscript we added an explanation that the nodal lines in Ca and those in Ag are both trivial in the sense of the Z_2 topological number in Ref. [6]. We also add calculation results to support this conclusion in Supplemental Materials.

6) We deleted the description of the value “1eV” from the text “... *increases the energy of* the p-orbital by about 1 eV ...”

7) We extensively changed the text so that the bulk physics and surface physics are clearly distinguished. The main point is that Berry phase and Zak phase are bulk quantities, and this bulk Zak phase is related with a surface polarization charge. In addition we added a detailed explanation on this relationship between bulk Zak phase and surface polarization charge, based on the paper by Vanderbilt and King-Smith (PRB48,4442 (1993) , ref.43), to explain that the value of polarization is obtained from the Zak phase.

8) we added this explanation in the caption of Fig.3 that the surface region here refers to one atomic layer near the surface.

9) In the caption Fig.3i, we added an explanation on the vertical axis “charge density”, that the charge density per 2D unit cell (along the surface) within each atomic layer, measured in the unit of the electronic charge e .

10) We added an explanation that the state that crosses the Fermi level between K and Gamma (see inset to 3g) is NOT a surface state. We also added a detailed explanation how this charge deficiency at $\forall \Lambda_1$ in Fig. 3i is explained as a bulk effect, and not as a surface state. We also noted the definition of the surface states adopted in the paper to show that there is a clear distinction between bulk states and surface states.

11) We removed the phrase “because the surface breaks inversion symmetry” in page 3 from the text.

12) We added a comment that even when the SOI is present and a gap opens at the nodal line, various effects of nodal lines remain, such as the drumhead surface states

and charge polarizations.

13) We have added a description on the lattice parameter, the k-integration method, and the size of the vacuum gap in the Methods section.

14) In the Supplemental Material, we added a calculation of the screening of the polarization by the bulk carriers based on the Poisson equation, and rough estimation of the resulting surface dipole density and the strength of the potential dip. In addition the effect of the potential onto the surface states, if any is discussed. In the main text we briefly show the results of these calculations and estimations.

15) We added the calculation on full self-consistent slab calculations in Supplemental Material to show the effect of lattice relaxation by the DFT/GGA and compare the result with that based on the Wannier function.

16) We thoroughly have checked the text and corrected the errors.

=====
=====
=====
=====

Reviewers' comments:

Reviewer #1 (Remarks to the Author):

Report on "Topological Dirac Nodal Lines in fcc Calcium, Strontium, and Ytterbium" by Motoaki Hirayama et al.

=====

This manuscript discusses an interesting observation, the presence of nodal lines in the bulk band structure of elemental alkaline earth metals. While the very existence of these nodal lines has been observed already 50 years ago (Refs. 34 and 35), the authors now try to establish a relation to modern notations in terms of Berry phase and topological origin.

I think that such a venture is timely and of interest for a broad

community, since the subject are elemental metals which can be prepared in much simpler and better reproducible way than, e.g., ternary or quaternary compounds which are in the focus of most of the current research. However, I do not trust that the results are correct - see below. For this reason I suggest rejection of the manuscript but would encourage the authors to carefully check the situation in order to find out, if there is a way to remedy their arguments.

Major problems:

- The authors write that there are two reasons for the emergence of nodal lines, (A) mirror symmetry and (B) topology. They refer to the Methods section. There, both cases are explained in a few sentences. While I immediately understand the reasons for case (A), case (B) is not clear to me from the given explanation. Either, a reference should be given, or, a clear mathematical proof is needed for this topology-related matter.

- Most of the slab calculations are done in 3D supercell geometry using Wannier functions. (This is not very explicitly stated, but from all what I see and understand, it is the case.) The use of Wannier functions poses a serious accuracy problem for the position of the surface states. The authors briefly discuss the problem when comparing Fig. 3a with Fig. 5b.

However, they arrive at the (in my understanding wrong) conclusion that the difference is due to the neglect of surface lattice relaxation in the Wannier approach. What matters much more, are two other problems:

(i) The level shift due to the unavoidable surface dipole. No effect of electrostatic shifts due to the surface is included in the usual Wannier approach. Thus, the energy position of states close to the surface is not accurate. From the comparison of the two figures one finds that this shift amounts to about 0.5 eV in the case of Ca.

(ii) The basis may not be complete close to the surface, see e.g. Ref. [43].

Thus, no firm conclusion about the surface states position can be drawn from these calculations, including all related implications. Full self-consistent slab calculations would be needed.

- The calculation of Berry phases should be applied to the *bulk* system. The authors, however, seem to perform the calculation for 3D supercells simulating the surface. This means, that the results do not apply, e.g., for Ca but for an artificial crystal consisting of Ca stacks, separated by vacuum.

This is my most important point.

Minor points:

- While most of the text is well understandable, there are a few spelling and grammar errors that should be corrected.

- Fig. 2: "... nonequivalent nodal lines ..." To my mind, all lines shown in Fig. 2b are equivalent by cubic symmetry, as well as all lines shown in Fig. 2g.

- Pg. 2: "... is almost flat near the L points, ..." Well, the dispersion is about 0.3 eV, not so small.

- Pg. 2: "It gives four nodal lines ..." I count eight. The same problem is present at several places throughout the manuscript.

- Pg. 2: "... increases the energy of the prorbital by about 1 eV ..." How is this number obtained?

- Fig. 3: "... in the surface region ..." should be quantified.

- Fig. 3: I would expect a different unit for a charge density.

- Fig. 3g and 3i: The state that crosses the Fermi level between K and Gamma (see inset to 3g) is clearly a surface state. This is obvious from Fig. 3i:

As soon as the state has crossed the Fermi level, charge is added to the two outermost layers on each side. I think that the problem is related with the (not quantified, see above) definition what a "surface state" is. Anyway, this observation makes a lengthy discussion on Pg. 3 invalid.

- Fig. 3g and 3h, discussion about the contribution of surface states to the polarization: The resolution is not very good, but I think that the jump of the Berry phase close to K (rightmost part of Fig. 3h) is related with the surface state crossing the Fermi level between M and K. Note: If you use 3D supercells, there is no fundamental difference between surface states and bulk states.

- Pg. 3: "... because the surface breaks inversion symmetry." I understand from other parts of your text, that you are using supercell with inversion center.
- Pg. 4: "We attribute this large splitting to the strong electric field from the nodal lines ..." This is something I would not undersign without a clear proof. The nodal lines may give a contribution (in k-space, but probably smeared in real space). This would have to be precisely quantified. Alternatively, did you check the case of Yb, were no nodal lines exist if spin-orbit coupling is considered?
- Methods: The k-integration method should be mentioned, not only the number of points.
- Methods: Which lattice parameter is used for the $p = 0$ calculations?
- Methods: Please give the size of the vacuum gap in the 3D supercells.
- Methods: Why is LDA used for the band calculations? Does GGA give any notable difference?
- Methods: The last remark is unclear. Fig. 5 shows nothing but electronic structure and lattice spacings. What has been calculated with Wannier functions in Fig. 5?

Reviewer #2 (Remarks to the Author):

In their manuscript titled "Topological Dirac Nodal Lines in fcc Calcium, Strontium, and Ytterbium", the authors study a class of fcc alkaline earth metals, exploring the possibility of having topological nodal lines in their band structure. They also analyze how the surface of such systems is affected by the presence of nodal lines, proposing two mechanisms by which the Rashba effect could be enhanced. The field of topological properties of band insulators and semimetals is nowadays very active, and in this respect the manuscript is timely and could be potentially appealing for quite a broad audience. Nonetheless, in my opinion the manuscript in its present form is not strong enough to be published in Nature Communications. The title suggests that topological nodal lines can be found in fcc Ca, Sr and Yb, while actually such nodal lines are present only when spin-orbit interaction (SOI) is neglected. Furthermore, a topological robust

nodal line is predicted only for Ca under pressure (again without SOI). The possible influence of nodal lines on surface Rashba effect seems to be more appealing, even though its importance and role in determining the strength of the Rashba splitting is in my opinion not fully supported by the present discussion. Overall, I found the manuscript not always clear and scientifically sound, as I try to elaborate in the following.

1) The authors differentiate between nodal lines emerging from topology or mirror symmetry. In my understanding, however, the origin of a nodal line always resides on the topological properties of the Bloch wave-functions in reciprocal space; as discussed in Ref [6], in systems without SOI the symmetry protection of nodal lines is provided by the composition T^*P of time-reversal (T) and parity (P) symmetries, whereas an additional mirror symmetry is required if a strong spin-orbit interaction is considered.

2) When discussing the nodal-line semimetal (NLS) properties of Ca, the authors say that the system can be classified as a weak NLS; then they say that Ca at ambient pressure is not a NLS, the latter phase being triggered by pressure. They discriminate between the two phases by looking (as far as I understand) at the curvature of the band dispersions around the L point. However, the classification proposed in Ref [6] is slightly different, and I would say more robust. First of all, the π -Berry phase (dubbed 1D Z_2 invariant) around nodal lines is a signature of the topological protection of the T^*P symmetry; this guarantees the presence of nodal lines but not its protection against perturbations (the system can be gapped by adiabatically tuning $H(k)$). The nodal line is topologically stable only if the 3D system possesses a Z_2 charge; quoting Ref. [6], "A nodal line with a Z_2 charge can be considered a Z_2 monopole, which can only be created or annihilated in pairs". In Ref. [6], a different 2D Z_2 invariant has been proposed to verify if a nodal-line semimetal possesses a Z_2 charge, classifying it as a topological (nodal-line) semimetal. Could the authors evaluate such 2D Z_2 invariants in order to support their conclusions?

3) In agreement with Ref. [6], neither Sr nor Yb present nodal lines, due to their non-negligible SOI. I wonder if the results shown for Ca still hold when SOI is included.

4) I'm really confused by the discussion of bulk polarization. The formula of the Berry phase (Eq. (1) in Methods) is formally analogous to the polarization of one-dimensional

systems; even in this case, however, the "real" polarization can be only defined by taking the difference of the Berry phase between two phases (the polar and the nonpolar ones) modulo a quantum. By no means the Berry phase of a single system can be related to its polarization; despite it can give informations about the charge distribution at the edges, I think that referring to it as a bulk polarization can be misleading. More technically, I do not fully understand how it is computed; is it calculated in slab geometry? Then the k_{\perp} is not defined. The authors should explicitly state if it has been calculated from the bulk model, in which case there is no breaking of inversion symmetry. On the other hand, if my understanding is correct, then it is not clear how such quantity has been evaluated in the case of depleted surfaces (Fig. 4 f) and g)).

5) There is some confusion between the Berry phase of loops around the nodal lines (that should be related to the 1D Z₂ invariant of Ref. [6]) and the Berry phase of the "polarization", i.e., calculated along k_{\perp} . The authors should clarify their relationship before relating the Berry phase to surface states and/or surface charges, further clarifying which Berry-phase are referring to. For instance, at page 3 right column, they say "in ref. 6 it is noted that the Berry phase does not imply existence of surface states because the surface breaks inversion symmetry"; but the Berry-phase discussed in Ref. 6 is that calculated along loops enclosing the nodal line, not the one given in Eq (1) in Methods.

6) When discussing how the nodal lines affect the surface, the authors say that nodal line affects the "bulk polarization" (a claim that I do not understand, giving my previous comments) and may additionally cause the appearance of surface states. Could the authors better clarify the relationship between these two effects? What is the origin of the surface states induced by the presence of bulk nodal lines?

7) The most interesting section of the manuscript is the suggestion that nodal lines can affect Rashba effects of a Bi monolayer on the fcc alkali metals. However, while the suggestion that surface charge redistribution (evaluated via the Berry-phase "polarization") and/or the presence of surface states of the substrate may enhance the Rashba effect of the Bi is indeed interesting, the informations provided in the manuscript are in my opinion not enough to support such claim. For instance, how large is the Rashba splitting of Bi monolayers on substrates without nodal lines? Can the authors estimate the enhancement of the effect with and without nodal lines? Furthermore, the authors claim that Ag substrate contribute to Rashba splitting mostly via hybridization

of Bi with emergent surface states arising from nodal lines; did they calculate the surface charge redistribution (using the Berry-phase "polarization"), in order to rule out its role?

Reply to Reviewer #1

First of all, we would like to thank the Reviewer #1 for his/her useful comments and suggestions. We revised the manuscript in the light of his/her comments, as described below.

Comment #1: - *Topological origin of the nodal lines (extending my first comment of the first review):*

The authors find, that the Zak phase around the nodal lines is π and conclude that this points to a topological origin. I am not sure, if this is justified. Could the nodal lines be due to a symmetry different from the mirror symmetry? The authors introduce the notation "weak" NLS, but before accepting a new notation, one has to be sure about its meaning.

Here, it seems to me that the causality is not clear: The nodal lines are there, and only after their identification the Zak phase can be calculated. So, is there any primary topological reason for their existence? If such a reason cannot be identified, one cannot talk about "topological origin".

Reply to Comment #1:

There is a primary topological reason for their existence. In the present case, at a single point on each of the L-W lines, the two bands cross because on the L-W line (i.e. C2 axis) they have different C2 eigenvalues. Then, because the three conditions (spinless, inversion symmetry, time-reversal symmetry) for existence of nodal lines is satisfied in this case, we can show that this degeneracy cannot be an isolated point in k-space, but extends along a curve in k-space, forming a nodal line, due to the π Berry phase. Thus the nodal lines in the present material comes from the topological origin.

As the Reviewer #1 pointed out, the causality for the existence of the nodal lines was not clear in the previous manuscript. Therefore we revised the explanations in the main text, in the Methods section and in the Supplemental Material to make the causality clearer. In particular, we added a section "Nodal lines stemming from π Berry phase" in the Supplemental Material to explain the details how the nodal lines originate from the mechanism of the π Berry phase.

Comment #2: - *In most of the text, the authors say that the surface polarization can be obtained by evaluating the area in k -parallel, where the Zak phase is not zero. Suddenly, on Pg. 5, this is abrogated by telling that also the (001) and (110) surfaces show a large polarization due to jumps of the phase by 2π . This means, there is no clear prediction possible about the nodal line impact on Rashba splitting: It is only one of several additive contributions, and not necessarily the dominating one.*

Reply to Comment #2:

This comment of the Reviewer #1 is indeed correct. Apart from the nodal line, there are various factors that contribute to the surface Rashba splitting. We cannot conclude the present mechanism from the nodal lines is the dominating one, and this point was not clear in the previous manuscript. In fact, the main point of the present paper is to show the relationship between the nodal lines and the surface charge. To better clarify this main conclusion we revised the text accordingly.

Comment #3: - *The grammar would need to be corrected by a native speaker.*

Reply to Comment #3: The manuscript has been checked and revised by two native speakers.

Comment #4: - *Parts of the discussion are hard to follow, not always with a clear logic, and with jumps in the argumentation.*

Reply to Comment #4: We revised several parts of the text to avoid jumps in the argumentation. For example, as is pointed out by Reviewer #2, it was not clear how the last paragraph of the section “Zak phase and surface states” is related to the main argument, so we revised this paragraph to make the logic smoother. Further revisions are made in various parts of the text for clearer logic.

Comment #5: - *Ref 14 is incomplete; Ref. 2 of the Supplement is wrong.*

Reply to Comment #5: We corrected the text accordingly.

Comment #6: - *Supplement, "wavefunction" should read "vector" before Eq. (8).*

Reply to Comment #6: We corrected the text accordingly.

Reply to Reviewer #2

First of all, we would like to thank the Reviewer #2 for his/her useful comments and suggestions. We revised the manuscript in the light of his/her comments, as described below.

Comment #1: *-The main result of the manuscript is in my opinion the relationship between the bulk nodal lines and the surface charges (rather than states), which could possibly have consequences on the surface Rashba splitting; the title, however, seems to me a bit out of focus, referring to topological nodal lines in elemental metals (which, by the way, are not so robust with respect to spin-orbit interaction, as also stated in the main text; on the other hand, the authors suggest that the bulk/surface correspondence should survive for moderate SOI even in the presence of small SOI-induced gap, implying that the latter, rather than the presence of topological nodal lines, is the meaningful result).*

Reply to Comment #1:

We agree with this comment by the Reviewer #2. The main focus of the paper is indeed the relationship between the bulk nodal lines and the surface charges. On the other hand, we also emphasize that this relationship is clearly seen in real and simple materials such as calcium. From these reasons, we revised the title from “topological Dirac nodal lines in fcc calcium, strontium and ytterbium” to “topological Dirac nodal lines and surface charges in fcc calcium, strontium and ytterbium”. We also revised the text to show that the main focus of the paper is the relationship between the bulk nodal lines and the surface charges.

Comment #2: *While I think that the manuscript readability has been significantly improved with respect to the definition of Berry phase and Zak phase, in order to avoid further misunderstandings I would suggest the authors to relate the Zak phase to a surface charge, rather than polarization, surface polarization or topological polarization (as in fact done in Ref 43)*

Reply to Comment #2:

We are grateful to the reviewer #2 for the helpful comment. We totally agree with this

comment. Our point is that the three quantities, Zak phase, polarization, and surface charge, are mutually related. Nevertheless, to avoid misunderstanding, it is better to directly relate the Zak phase to a surface charge directly, as pointed out by the reviewer #2.

For example, within the region with π Zak phase, the 1D polarization is $e/2 \pmod{e}$ and surface charge density (per unit cell) is also $e/2 \pmod{e}$. Nevertheless, because it is a centrosymmetric crystal, it might be misleading to say that it has a nonzero polarization. In the present case, the physical consequence of the π Zak phase is the surface charges, and therefore, we revised the text to directly relate the Zak phase with a surface charge.

Comment #3: -As it stands, the conclusive paragraph of the section titled "Zak phase and surface states", where surface depletion is considered, seems unrelated to the previous discussion, being unclear what conclusion can be drawn from the numerical results. Is it the robustness of Zak phase/ surface charges? I would suggest the authors to add a clear conclusive sentence at the end of the paragraph also and not only in the summary section.

Reply to Comment #3:

It is indeed true that this paragraph on the surface depletion seems unrelated to the previous discussion. Therefore we revised the paragraph to clarify the relationship with the previous context, by explaining that the surface charge (coming from the Zak phase) is a bulk quantity (modulo e) and is unaffected by surface depletion.

Comment #4: Eventually, while the proposed mechanism (surface charge and reduced screening due to the presence of nodal lines, leading to an increase of the surface electric field) for enhanced surface Rashba splitting is reasonable, there is no clear evidence of its prominence, since several other factors are at play which can be hardly ruled out (as admitted by the authors themselves). Even though I believe it is an interesting proposal, I would suggest the authors to smooth their claims about the enhancement of Rashba splitting as due to the presence of nodal lines near the Fermi level, clarifying that it is, in fact, a speculation.

Reply to Comment #4:

We agree with this comment by the Reviewer #2. Therefore, we revised the text so that

the nodal line near the Fermi level is only among various factors which gives surface Rashba splitting. We tried to evaluate the amount of enhancement of Rashba splitting from the nodal lines, but it is not possible since various factors are entangled in the Rashba splitting, and one cannot separate the effect of the nodal lines, as discussed in the manuscript.

List of corrections

1) We revised the explanations in the main text, in the Methods section and in the Supplemental Material to clarify that the nodal lines in the present materials emerges from the topological reason (π Berry phase). In particular, we added a section “Nodal lines stemming from π Berry phase” in the Supplemental Material to explain the details how the nodal lines originate from the mechanism of the π Berry phase.

2) Throughout the paper, the grammar has been corrected by two native speakers.

3) We revised the last paragraph of the section "*Zak phase and surface states*" to clarify the relationship with the previous context, by explaining that the surface charge (coming from the Zak phase) is a bulk quantity (modulo e) and is unaffected by surface depletion.

4) We corrected the references (Ref 14 in the main text and Ref. 2 of the Supplement).

5) We revised the title from “topological Dirac nodal lines in fcc calcium, strontium and ytterbium” to “topological Dirac nodal lines and surface charges in fcc calcium, strontium and ytterbium”.

6) We revised the text thoroughly, in order to directly relate the Zak phase with a surface charge, without relating it to bulk polarization.

7) We revised the discussion on possible enhancement of the Rashba splitting, to show that the contribution of the nodal lines is among various contributions to the Rashba splitting.

Reply to the Reviewers:

We thank the reviewers for careful reading and helpful comments. In the light of Reviewer #1's comment, we corrected Ref. [S1], i.e. the year is changed from 2982 to 1982.

-----Reviewers' Comments--

Reviewer #1 (Remarks to the Author):

Most remarks of my previous report were considered by the authors and related necessary changes to the manuscript were implemented. There is one technical point: Ref. [S1] (previous Ref. [S2]) is still wrong.

Reviewer #2 (Remarks to the Author):

I believe that the authors have replied to all comments and issues previously raised in a satisfactory way. I have no further comments and I think that the manuscript is suitable for publication in Nature Communications